# Leveraging electronic health records from two hospital systems identifies male infertility-associated comorbidities across time

Sarah R. Woldemariam [1], Feng Xie[2,3,4,5], Alennie Roldan [1], Jacquelyn Roger [1], Alice S. Tang [1], Tomiko T. Oskotsky [1,6], David K. Stevenson[3], Ruth B. Lathi[7], Aleksandar Rajkovic [8,9,10], Isabel E. Allen[11], Nima Aghaeepour [2,3,4], Michael Eisenberg[12] ✉ & Marina Sirota [1] ✉

## Abstract

**Background** Male infertility (MI) is the sole cause of 20–30% of infertility cases, and it is a contributing factor for an additional 15–20% of cases. However, the full breadth of potential MI risk factors and adverse health outcomes has not been explored.

**Methods** We used electronic health records (EHRs) from the University of California (UC) and Stanford to identify MI-associated comorbidities. We identified 6531 and 5551 MI patients at UC and Stanford, respectively, and 8353 and 2464 vasectomy control patients at UC and Stanford, respectively. Low-dimensional embeddings of patients' diagnosis profiles based on MI status, demographics, or hospital utilization were compared using either Kruskal–Wallis tests followed by post hoc Dunn's tests or Mann–Whitney $U$ tests. We used logistic regression to identify MI-associated comorbidities prior to or after 6 months of a patient's first MI or vasectomy-related record. Pearson correlation coefficients were used to compare primary versus sensitivity logistic regression analyses as well as UC versus Stanford logistic regression analyses. Cox regression was used to assess whether patients had a higher risk of receiving diagnoses significantly associated with MI after the 6-month cutoff at UC.

**Results** Here, we identify 15 diagnoses that are positively associated with MI before the 6-month cutoff across both hospital systems and all analyses, including less expected comorbidities such as hypothyroidism and other anemias. Using Cox regression, we find that patients have a higher risk of receiving 11 out of 13 diagnoses positively associated with MI after the 6-month cutoff at UC.

**Conclusions** Our findings can set the groundwork for future studies to clarify the relationship between less expected comorbidities and MI.

## Plain Language Summary

15% of those attempting to conceive experience infertility. Male infertility (MI) is the sole cause of 30% of infertility cases and a contributory cause for an additional 15–20% of cases. The causes underlying MI are often unknown, and risk factors for and adverse outcomes of MI have been primarily studied in the context of specific conditions. Here, we use electronic health records from the University of California and Stanford University to identify associated conditions that may be risk factors for or adverse outcomes of MI. We found expected conditions such as varicose veins as well as unexpected conditions such as hypothyroidism. Our findings lay the foundation for future work that could clarify the relationship between these conditions and MI, potentially leading to new avenues for its prevention and treatment.

Infertility affects 15% of those attempting to conceive after one year of unprotected regular intercourse[1]. Male infertility (MI) is the sole contributory factor for 20–30% of infertility cases and a partial contributory factor for an additional 20% of cases[2,3]. The etiology of MI is wide-ranging and could be attributed to genetic causes such as CFTR mutations, acquired causes such as gonadotoxin exposure, or, in the case of 30-50% of male factor infertility cases, idiopathic[2]. Moreover, MI may be poised to become a

major concern in the future, as studies have shown that sperm counts have been decreasing at an accelerating pace worldwide for decades[4]. Despite this, male factor infertility is relatively understudied compared to female factor infertility[5].

There has also been growing consensus that MI could be a marker of current and future health[6–8]. Indeed, numerous retrospective and prospective case-control and cohort studies have investigated MI in the context

of numerous health outcomes, including whether MI is associated with an increased incidence of diabetes, cardiovascular disease, and autoimmune disease, as well as whether conditions such as chronic kidney disease could be a risk factor for MI[9-13]. However, there have been no prior studies leveraging a data-driven approach for identifying the full breadth of potential comorbidities that are risk factors for or adverse health outcomes of MI. Data-driven approaches leveraging electronic health records (EHR) in particular have contributed to new insights into complex conditions, such as Alzheimer's disease, recurrent pregnancy loss, and neonatal outcomes[14-18]. Additionally, findings validated across different hospital EHR systems (e.g., public and private) may highlight comorbidities that may be of particular clinical interest and relevance. A data-driven approach utilizing EHR may therefore lead to a more complete understanding of MI in the context of a patient's overall health before and after a MI diagnosis, shed light on the idiopathic causes of MI, and inspire new approaches to preventing and treating MI in patients.

Here, we perform an observational case-control study utilizing structured EHR to identify MI-associated comorbidities retrospectively and prospectively. The cases and controls in our study were nested and compared within and between two independent hospital systems to identify these associations. We explore conditions first diagnosed before 6 months after a MI diagnosis or vasectomy-related record to identify comorbidities that may have been present prior to a MI diagnosis; thus, these associated comorbidities may be risk factors for MI. We also explore conditions first diagnosed at least 6 months after a MI diagnosis or vasectomy-related record to identify comorbidities that may have manifested after a MI diagnosis; thus, these associated comorbidities may be adverse outcomes of MI. We then use Cox proportional hazards models to assess whether MI patients have a higher risk of receiving diagnoses found to be significantly associated with MI at UC after the 6-month cutoff. We find both expected and less-expected comorbidities; these less-expected comorbidities include hypothyroidism, other anemias, and prostate cancer, which are all associated with MI before the 6-month cutoff. Future longitudinal studies can clarify the relationship between these comorbidities and MI, which may ultimately lead to improved treatment and prevention strategies for MI.

## Methods

This study has been approved by the Institutional Review Boards of the University of California, San Francisco (#17-22929) and Stanford University (#39225). Written informed consent for this study was waived due to the utilization of de-identified UC and Stanford EHR databases.

### Patient selection

University of California (UC) patients were identified from the University of California Data Discovery Portal (UCDDP), a HIPAA-compliant limited data set of over 8 million patients' health records, from January 1, 2012 through June 30, 2023. Patient selection is summarized in Supplementary Fig. 1.

We identified male-identified MI patients (cases) using MI OMOP concept ids (Supplementary Data 1). These male-identified patients were identified using the OMOP field gender_concept_id = MALE. While biological sex and gender are distinct, in OMOP, gender_concept_id refers to biological sex (https://github.com/OHDSI/Vocabulary-v5.0/wiki/Vocab.-Gender). We then filtered out patients with a vasectomy-related record using CPT-4, LOINC, or SNOMED corresponding OMOP concept ids (Supplementary Data 1). Next, we filtered for MI patients with at least one non-MI diagnosis (refer to *Mapping ICD diagnoses to phecode-corresponding phenotypes* section for details). Finally, we filtered for patients with an estimated age of at least 18; who received care at UC medical center 1, 2, 3, 4, or 5; had at least one associated area deprivation index (ADI); and received a MI diagnosis at least 6 months prior to the last day clinical information was recorded in the EHR (Supplementary Fig. 1).

We identified male-identified vasectomy patients (controls) using vasectomy-related CPT-4, LOINC, or SNOMED corresponding OMOP concept IDs (Supplementary Data 1). We then filtered out patients with a

MI diagnosis using MI OMOP concept ids (Supplementary Data 1). Next, we filtered for patients with at least one phecode-corresponding phenotype (refer to *Mapping ICD diagnoses to phecode-corresponding phenotypes* section for details). Finally, we filtered for patients with an estimated age of at least 18; who received care at UC medical center 1, 2, 3, 4, or 5; had at least one associated ADI; and had their first vasectomy-related record at least 6 months prior to the last day clinical information was recorded in the EHR.

### Mapping ICD diagnoses to phecode-corresponding phenotypes

The diagnoses utilized in our study are phecode-corresponding phenotypes. We obtained UC patients' ICD-9-CM and ICD-10-CM diagnoses from UCDDP. We then mapped these ICD diagnoses to phecode-corresponding phenotypes using the Phecode Map 1.2 with ICD-9 Codes table[19,20] for ICD-9-CM diagnoses or the Phecode Map 1.2 with ICD-10 Codes table for ICD-10-CM diagnoses to enable the aggregation and categorization of diagnoses[21]. We then identified whether a patient was first diagnosed with a given phenotype prior to or beyond 6 months after their first MI diagnosis or vasectomy-related record.

### Patient visualization using dimensionality reduction

We used Uniform Manifold Approximation and Projection (UMAP) to perform low-dimensional embedding of patients' diagnosis profiles using the Python package umap-learn. Low-dimensional embedding takes as input an $m \times n$ matrix, where each row ($m_i$) represents a patient, and each column ($n_j$) represents a particular diagnosis. The column values indicate whether the patient has or does not have the particular diagnosis using the binary values 1 or 0, respectively. UMAP outputs an $m \times 2$ matrix from the $m \times n$ matrix, which allows for the visualization of patients' diagnosis profiles in two dimensions. We performed low-dimensional embedding on patients' diagnosis profiles of diagnoses first obtained at any time; of diagnoses first obtained prior to 6 months after patients' first MI diagnosis or vasectomy-related record; and of diagnoses first obtained beyond 6 months after patients' first MI diagnosis or vasectomy-related record. UMAP distributions were tested for significance using either Mann–Whitney $U$ tests (for comparing two categories) or Kruskal–Wallis tests followed by two-sided Dunn's tests with Bonferroni correction (for comparing more than two categories).

### Association analysis

We used logistic regression to identify diagnoses positively or negatively associated with MI, stratified by whether diagnoses were first diagnosed prior to or beyond 6 months after patients' first MI diagnosis (cases) or vasectomy-related record (controls). We ran a primary analysis as well as two sensitivity analyses on the stratified data for a total of 6 analyses. For each analysis, a given diagnosis must be diagnosed in at least 1 case and 1 control patient.

Three analyses were performed for diagnoses first diagnosed prior to 6 months after the patients' first MI diagnosis or vasectomy-related record. The primary analysis tested the association of MI and a given diagnosis adjusted by estimated age of the patient when they received their first male infertility diagnosis or vasectomy-related record and UC location. The social determinants of health (SDoH) sensitivity analysis tested the association of MI and a given diagnosis adjusted by estimated age of the patient when they received their first male infertility diagnosis or vasectomy-related record, UC location, self- or provider-identified race, self- or provider-identified ethnicity, and ADI. The hospital utilization sensitivity analysis tested the association of MI and a given diagnosis adjusted by estimated age of the patient when they received their first male infertility diagnosis or vasectomy-related record, UC location, number of visits prior to 6 months after patients' first MI diagnosis or vasectomy-related record, and months in the EHR prior to 6 months after patients' first MI diagnosis or vasectomy-related record.

The other analyses were performed for diagnoses first diagnosed beyond 6 months after the patients' first MI diagnosis or vasectomy-related record. The primary analysis tested the association of MI and a given

diagnosis adjusted by estimated age of the the patient when they received their first male infertility diagnosis or vasectomy-related record and UC location. The SDoH sensitivity analysis tested the association of MI and a given diagnosis adjusted by estimated age of the patient when they received their first male infertility diagnosis or vasectomy-related record, UC location, self- or provider-identified race, self- or provider-identified ethnicity, and ADI. The hospital utilization sensitivity analysis tested the association of MI and a given diagnosis adjusted by estimated age of the patient when they received their first male infertility diagnosis or vasectomy-related record, UC location, number of visits beyond 6 months after patients' first MI diagnosis or vasectomy-related record, and months in the EHR beyond 6 months after patients' first MI diagnosis or vasectomy-related record.

We ran the above three analyses for patients with no fixed follow-up time, as well as for patients with at least 12, 24, 36, 48, or 60 months of follow-up time. For fixed follow-up times, we explored diagnoses obtained within those timeframes.

The number of months patients were in the EHR was measured by subtracting the last visit date from the first visit date before or after the 6-month cutoff. We used Benjamini-Hochberg correction to identify diagnoses significantly associated with MI. We visualized primary analysis results using volcano and Manhattan plots. We also used ln-ln plots to visually compare the significance and odds ratios of diagnoses between the primary analyses and SDoH sensitivity analyses, between the primary analyses and hospital utilization sensitivity analyses, and between primary analyses of diagnosis associations prior to versus beyond 6 months after patients' first MI diagnosis or vasectomy-related record. Finally, we used upset plots to visually compare significant diagnoses across combinations of the primary and sensitivity analyses.

### Cox proportional hazards models

We used Cox proportional hazards models to assess whether patients with an MI diagnosis (cases) had a higher risk of receiving MI-associated diagnoses relative to patients with a vasectomy-related record (controls). We focused on diagnoses that were first received beyond 6 months after a patient's first MI diagnosis and that were significantly associated with MI at UC for all 3 logistic regression analyses for a total of 13 diagnoses. For each analysis, a given diagnosis must be diagnosed in at least 1 case and 1 control patient.

Three analyses were performed for diagnoses first diagnosed beyond 6 months after the patients' first MI diagnosis or vasectomy-related record. The primary analysis tested the risk of receiving a given diagnosis based on having MI, adjusted by the estimated age of the patient when they received their first male infertility diagnosis or vasectomy-related record and UC location. SDoH sensitivity analysis tested the risk of receiving a given diagnosis based on having MI, adjusted by estimated age of patient when they received their first male infertility diagnosis or vasectomy-related record, UC location, self- or provider-identified race, self- or provider-identified ethnicity, and ADI. The hospital utilization sensitivity analysis tested the risk of receiving a given diagnosis based on having MI, adjusted by estimated age of patient when they received their first male infertility diagnosis or vasectomy-related record, UC location, number of visits beyond 6 months after patients' first MI diagnosis or vasectomy-related record, and months in the EHR beyond 6 months after patients' first MI diagnosis or vasectomy-related record.

EHR cutoff date was used to measure time-to-event for patients who did not receive a given diagnosis of interest; this date corresponds to June 30, 2023, the date the EHR was last refreshed for this study. Time-to-event corresponds to the number of days between patients' first male infertility diagnosis (cases) or vasectomy-related record (controls) and patients' first diagnosis of the diagnosis of interest.

### External validation

For external validation, Stanford patients with MI were identified from the Stanford de-identified EHR containing patient records from over 3.7 million patients from 1999 through July 16, 2023. Patient selection is summarized in Supplementary Fig. 2 using the OMOP concept codes in Supplementary Data 1 for identifying patients with MI or those who have had a vasectomy. The main difference in selection criteria for Stanford patients is that the Stanford EHR does not record ADIs for their patients. With the exception of Cox proportional hazards models, Stanford patients underwent the same analyses as UC patients, including low-dimensional embeddings of their diagnosis profiles as well as association analyses using logistic regression.

Three association analyses were performed for diagnoses first obtained prior to 6 months after patients' first MI diagnosis or vasectomy-related record at Stanford. The primary analysis tested the association of MI and a given diagnosis adjusted by estimated age of the patient when they received their first male infertility diagnosis or vasectomy-related record. The SDoH sensitivity analysis tested the association of MI and a given diagnosis adjusted by estimated age of the patient when they received their first male infertility diagnosis or vasectomy-related record, self- or provider-identified race, and self- or provider-identified ethnicity. The hospital utilization sensitivity analysis tested the association of MI and a given diagnosis adjusted by estimated age of the patient when they received their first male infertility diagnosis or vasectomy-related record, number of visits prior to 6 months after patients' first MI diagnosis or vasectomy-related record, and months in the EHR prior to 6 months after patients' first MI diagnosis or vasectomy-related record.

Association analyses were also performed for diagnoses first obtained beyond 6 months after patients' first MI diagnosis or vasectomy-related record. The primary analysis tested the association of MI and a given diagnosis adjusted by estimated age of the patient when they received their first male infertility diagnosis or vasectomy-related record. The SDoH sensitivity analysis tested the association of MI and a given diagnosis adjusted by estimated age of the patient when they received their first male infertility diagnosis or vasectomy-related record, self- or provider-identified race, and self- or provider-identified ethnicity. The hospital utilization sensitivity analysis tested the association of MI and a given diagnosis adjusted by estimated age of the patient when they received their first male infertility diagnosis or vasectomy-related record, number of visits beyond 6 months after patients' first MI diagnosis or vasectomy-related record, and months in the EHR beyond 6 months after patients' first MI diagnosis or vasectomy-related record.

As with UC, we ran the above three analyses for patients with no fixed follow-up time, as well as for patients with at least 12, 24, 36, 48, or 60 months of follow-up time. For fixed follow-up times, we explored diagnoses obtained within those timeframes.

Like UC, results for the Stanford association analyses were visualized using volcano plots and Manhattan plots and compared using ln-ln plots and upset plots. Temporally-stratified primary association analysis results from UC and Stanford were compared using ln-ln plots and upset plots.

### Statistics and Reproducibility

Unless otherwise noted, all packages utilized were Python packages. Patient information from EHR databases were queried using SQL. Patient filtering, data preprocessing, and data analysis were performed using pandas. Patient demographics tables were created using the R package tableone. UMAP was performed using umap-learn. SciPy was used for Kruskal-Wallis tests, Mann-Whitney $U$ tests, and obtaining Pearson correlation coefficients, while scikit-posthocs was used for post hoc Dunn's tests. Logistic regression analyses were performed using statsmodels. Ln-ln, volcano, and Manhattan plots were visualized using matplotlib and seaborn, while upset plots were visualized using the package UpSetPlot. Cox regression analyses were performed using the coxph function from the R package survival, with Kaplan-Meier survival curves visualized using the ggsurvplot function from the R package survminer. Large language models were utilized to aid in writing a portion of the Cox regression code. We identified 6531 patients with a MI diagnosis and 8353 patients with a vasectomy-related record at UC, and we identified 5551 patients with a MI diagnosis and 2464 patients with a vasectomy-related record at Stanford. We suggest similar patient numbers for reproducibility.

## Table 1 | UC demographics

| Covariate | Vasectomy 8353 patients | Male Infertility 6531 patients | p-value |
|---|---|---|---|
| Estimated age, mean (sd) | 45.57 (7.67) | 43.97 (8.77) | <0.001 |
| Estimated age at diagnosis or procedure, mean (sd) | 41.20 (7.14) | 38.45 (8.18) | <0.001 |
| Location, n (%) | | | <0.001 |
| 1 | 691 (8.3) | 615 (9.4) | |
| 2 | 1418 (17.0) | 1119 (17.1) | |
| 3 | 1670 (20.0) | 1018 (15.6) | |
| 4 | 1235 (14.8) | 2174 (33.3) | |
| 5 | 3339 (40.0) | 1605 (24.6) | |
| Identified race, n (%) | | | <0.001 |
| American Indian or Alaska Native | 32 (0.4) | 20 (0.3) | |
| Asian | 474 (5.7) | 925 (14.2) | |
| Black or African American | 250 (3.0) | 273 (4.2) | |
| Multirace | 254 (3.0) | 175 (2.7) | |
| Native Hawaiian or Other Pacific Islander | 24 (0.3) | 40 (0.6) | |
| Other Race | 798 (9.6) | 812 (12.4) | |
| Unknown | 1519 (18.2) | 1124 (17.2) | |
| White | 5002 (59.9) | 3162 (48.4) | |
| identified ethnicity, n (%) | | | 0.02 |
| Hispanic or Latino | 1087 (13.0) | 797 (12.2) | |
| Not Hispanic or Latino | 6138 (73.5) | 4755 (72.8) | |
| Unknown | 1128 (13.5) | 979 (15.0) | |
| ADI, mean (sd) | 3.63 (2.40) | 3.78 (2.56) | <0.001 |
| Number of visits before, mean (sd) | 24.87 (32.69) | 23.27 (45.05) | 0.012 |
| Number of visits after, mean (sd) | 16.94 (32.81) | 22.82 (52.48) | <0.001 |
| Months in EHR before, mean (sd) | 37.48 (33.44) | 22.16 (27.66) | <0.001 |
| Months in EHR after, mean (sd) | 20.66 (27.18) | 23.09 (30.67) | <0.001 |

The chi-squared test was used to calculate p-values for categorical variables. Welch's t test was used to calculate p-values for continuous variables.

*n* number of patients, *sd* standard deviation, *ADI* area deprivation index, *EHR* electronic health record, *number of visits before* number of visits before the 6-month cutoff, *number of visits after* number of visits after the 6-month cutoff, *months in EHR before* months in EHR before the 6-month cutoff (calculated as length of time between patient's last visit and patient's first visit before the 6-month cutoff), *months in EHR after* months in EHR after the 6-month cutoff (calculated as length of time between patient's last visit and patient's first visit after the 6-month cutoff).

### Reporting summary

Further information on research design is available in the Nature Portfolio Reporting Summary linked to this article.

## Results

Across five medical centers from the University of California (UC), we identified 6531 patients with a MI diagnosis (mean age 43.97; sd 8.77) and 8353 patients with a vasectomy-related record (mean age 45.57; sd 7.67) from 2012 to 2023. From Stanford, we identified 5551 patients with a MI diagnosis (mean age 46.92; sd 8.85) and 2,464 patients with a vasectomy-related record (mean age 46.27; sd 7.75) from 1999 to 2023. We chose patients with a vasectomy-related record as controls since previous studies suggest that they are demographically similar to patients who seek treatment for MI and are

generally considered fertile[12]. UC and Stanford demographic characteristics are summarized in Table 1 and Supplementary Data 2, respectively. An overview of the study design and association analyses is described in Fig. 1.

We temporally stratified our analyses to identify associated comorbidities that may be more likely to be risk factors for (before the 6-month cutoff) or adverse health outcomes due to (after the 6-month cutoff) MI. We chose 6 months after a patient's first MI diagnosis or vasectomy-related record as the cutoff to account for conditions that may have developed prior to or concomitant with the development of MI. Diagnoses in our study were represented as phecode-corresponding phenotypes (see *Mapping ICD diagnoses to phecode-corresponding phenotypes* in Methods).

### Low-dimensional embedding reveals that patients' diagnosis profiles differ based on male infertility status, demographic features, and hospital utilization

First, we used Uniform Manifold Approximation and Projection (UMAP) to perform low-dimensional embedding of patients' diagnosis profiles to explore whether they differed based on MI status and other patient characteristics (see *Patient visualization using dimensionality reduction* in Methods for details). We performed three low-dimensional embeddings of patients' diagnosis profiles: of diagnoses first obtained at any time (UC: n = 14,884 patients and n = 1664 diagnoses as features; Stanford: n = 8015 patients and 1574 diagnoses as features), diagnoses first obtained prior to 6 months of patients' first MI diagnosis or vasectomy-related record (i.e., before the 6-month cutoff) (UC: n = 14,812 patients and n = 1614 diagnoses as features; Stanford: n = 7723 patients and n = 1493 diagnoses as features), and diagnoses first obtained beyond 6 months after patients' first MI diagnosis or vasectomy-related record (i.e., after the 6-month cutoff) (UC: n = 9234 patients and n = 1562 diagnoses as features; Stanford: n = 5564 patients and n = 1452 diagnoses as features).

We explored whether there were differences in UMAP component distributions based on MI status and found significant differences for the majority of comparisons, including for diagnoses first obtained at any time at UC (Supplementary Fig. 3a) and Stanford (Supplementary Fig. 3b); diagnoses first obtained before the 6-month cutoff at UC for both components (Fig. 2a) and Stanford for the first component (Fig. 2b); and diagnoses first obtained after the 6-month cutoff at UC (Fig. 2c). No significant differences were found for diagnoses first obtained after the 6-month cutoff at Stanford (Fig. 2d).

Second, we explored whether there were differences in UC patients' diagnosis profiles first obtained at any time based on estimated age, UC location, identified race, identified ethnicity, and ADI, a 1–10 index of neighborhood advantage, with 1 indicating the highest advantage[22]. We also explored differences in Stanford patients' diagnosis profiles based on estimated age, identified race, and identified ethnicity. Using Kruskal–Wallis tests, we found significant differences in UMAP component distributions for all features, with post hoc Dunn's tests revealing a number of significant pairwise comparisons (Supplementary Figs. 4,5, Supplementary Data 3–5).

Third, we explored whether patients' diagnosis profiles differed based on hospital utilization measures, specifically the number of visits and months in the EHR, before and after the 6-month cutoff at UC and Stanford. The number of visits and months in the EHR were grouped into quintiles and compared. Using Kruskal-Wallis tests, we found significant differences in UMAP component distributions for both hospital utilization measures at UC and Stanford, for both before and after the 6-month cutoff, with post hoc Dunn's tests revealing a number of significant pairwise comparisons (Supplementary Figs. 6,7, Supplementary Data 3–5).

We also explored outlier patient clusters in the UC and Stanford UMAPs before the 6-month cutoff and found that these patients, overall, had fewer diagnoses relative to patients who were not in the outlier cluster for both UC and Stanford (Supplementary Data 6).

Overall, from low-dimensional embedding, we found that patients' diagnosis profiles differed significantly based not only on MI status, but also by estimated age, location, identified race, identified ethnicity, ADI, and hospital utilization. Therefore, we ran association analyses that adjusted for these covariates.

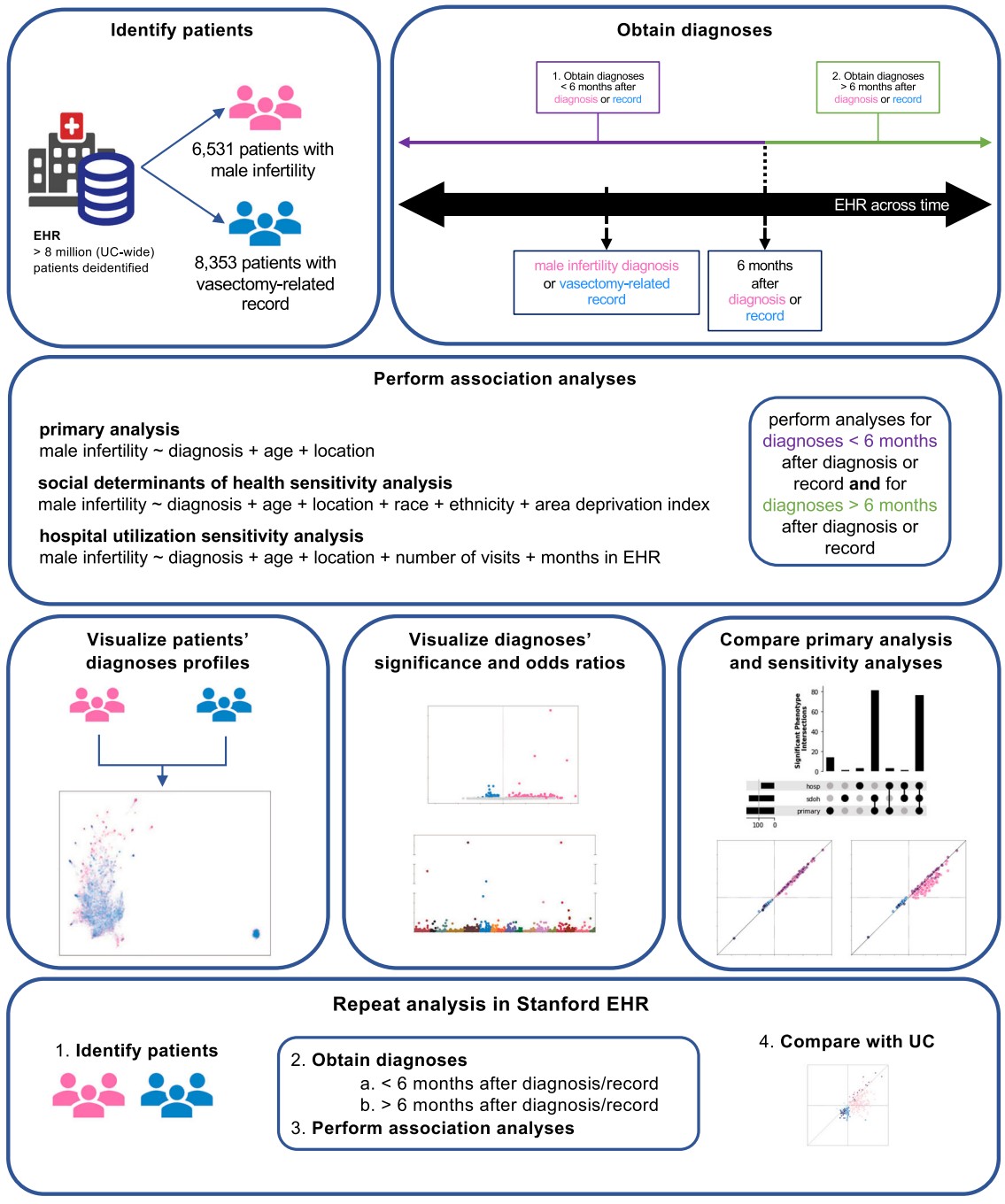

**Fig. 1 | Overview of association analyses.** Overview of patient identification, logistic regression analyses, visualization of results, and validation analysis. EHR = electronic health records. UC = University of California.

## Association analyses of diagnoses first diagnosed prior to 6 months after first male infertility diagnosis or vasectomy-related record reveal expected and less expected comorbidities across hospital systems

We used logistic regression to identify MI-associated diagnoses that were first obtained prior to 6 months after patients' first MI diagnosis or vasectomy-related record. For the UC primary analysis and both sensitivity analyses, we adjusted for the estimated age of the patient when they first received their MI diagnosis or vasectomy-related record as well as UC location. For the Stanford primary analysis and both sensitivity analyses, we adjusted for the estimated age of the patient when they first received their MI diagnosis or vasectomy-related record. The SDoH sensitivity analysis identified MI-associated comorbidities after adjusting for SDoH. For the UC

SDoH sensitivity analysis, we additionally adjusted for self- or provider-identified race, self- or provider-identified ethnicity, and ADI. For the Stanford SDoH sensitivity analysis, we additionally adjusted for self- or provider-identified race and self- or provider-identified ethnicity. Finally, the hospital utilization sensitivity analysis for UC and Stanford identified diagnosis associations with MI by additionally adjusting for the number of visits and months in the EHR before and after the 6-month cutoff (see Methods). For each analysis, we used Benjamini-Hochberg correction to identify significantly associated diagnoses. The distribution of timing between each diagnosis and the first MI diagnosis or vasectomy-related record can be found in Supplementary Data 7.

We tested 1442 diagnoses in each UC analysis to identify which comorbidities were significantly associated with MI. From the UC primary

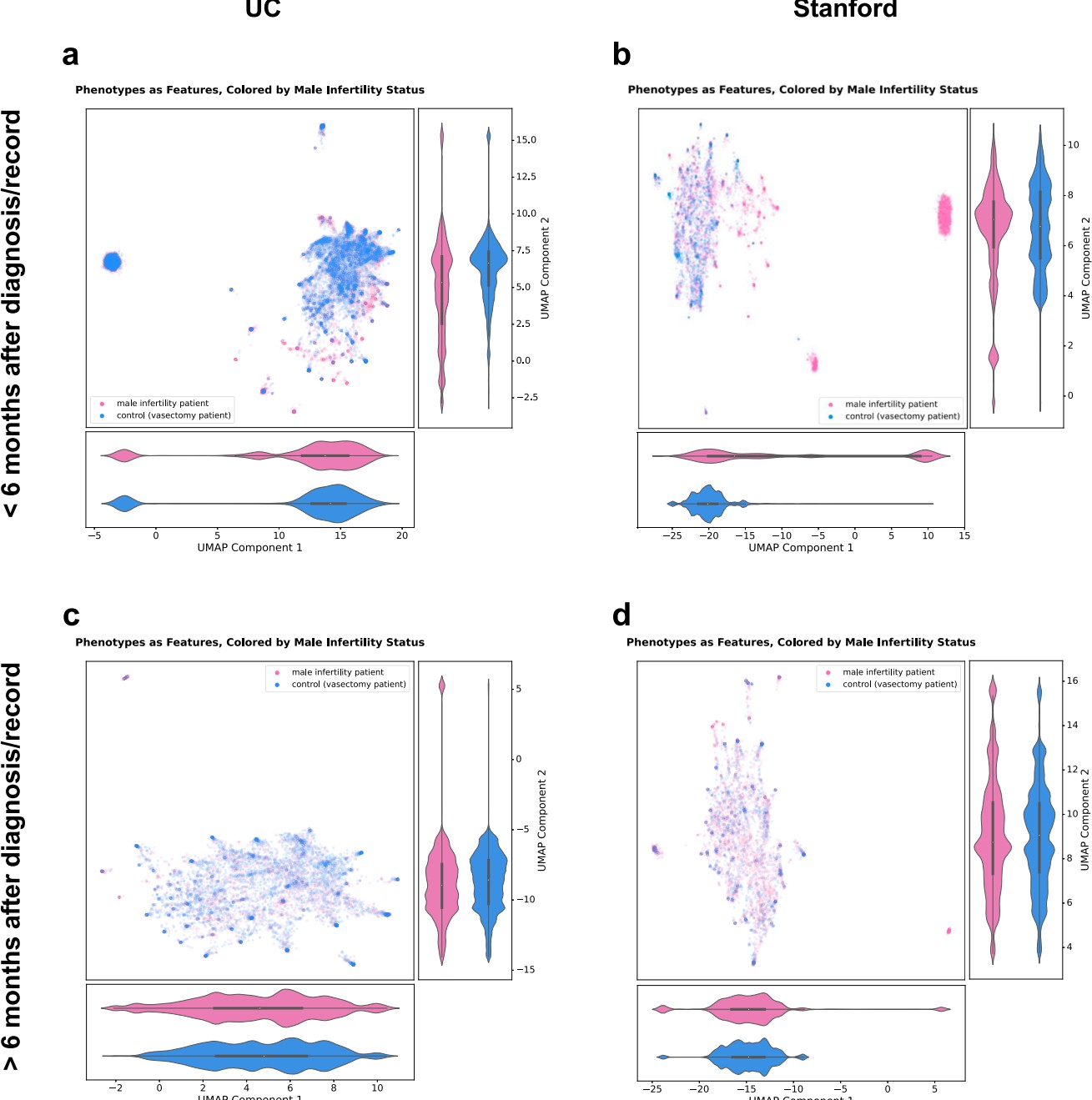

**Fig. 2 | Temporally stratified low-dimensional embedding of patients' diagnosis profiles reveals some separation based on patients' male infertility status.**
**a** UMAP of patients' diagnosis profiles for diagnoses first obtained before the 6-month cutoff at UC, colored by male infertility status ($n$ = 14,812 patients; 1614 diagnoses as features). **b** UMAP of patients' diagnosis profiles for diagnoses first obtained before the 6-month cutoff at Stanford, colored by male infertility status ($n$ = 7723 patients; 1493 diagnoses as features). **c** UMAP of patients' diagnosis profiles for diagnoses first obtained after the 6-month cutoff at UC, colored by male infertility status ($n$ = 9234 patients; 1562 diagnoses as features). **d** UMAP of patients' diagnosis profiles for diagnoses first obtained after the 6-month cutoff at Stanford,

colored by male infertility status ($n$ = 5564 patients; 1452 diagnoses as features). For each panel, the bottom violin plot shows distribution of UMAP component 1 based on male infertility status; the right violin plot shows distribution of UMAP component 2 based on male infertility status. Mann–Whitney $U$ tests assessed whether UMAP components significantly differed based on male infertility status, with a significance threshold of $p$-value < 0.05. Phenotypes correspond to diagnoses. Pink dots—patients with male infertility, blue dots—patients with a vasectomy-related record, *UMAP*—Uniform Manifold Approximation and Projection, *UC* University of California.

analysis, we found 205 positive and 63 negative associations (Fig. 3a, c, Supplementary Data 8). We also tested whether 1197 diagnoses in each Stanford analysis were significantly associated with MI. From the Stanford primary analysis, we found 19 positive and 106 negative associations (Fig. 3b, d, Supplementary Data 9).

From the UC SDoH sensitivity analysis, we found 190 positive and 55 negative associations, and from the UC hospital utilization sensitivity analysis, we found 79 positive and 32 negative associations (Supplementary Data 8). From the Stanford SDoH sensitivity analysis, we found 24 positive and 71 negative associations, and from the Stanford hospital utilization

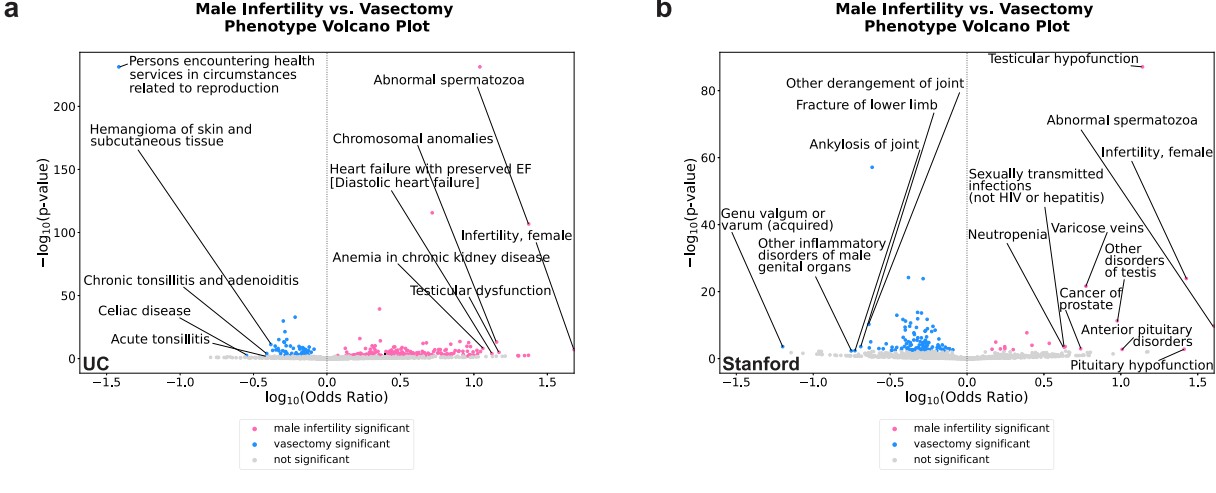

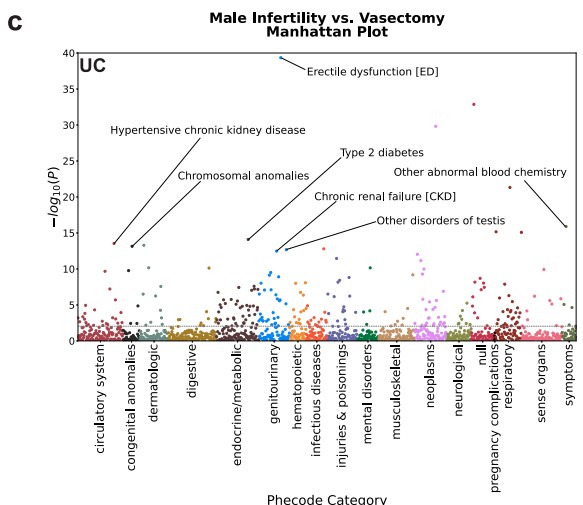

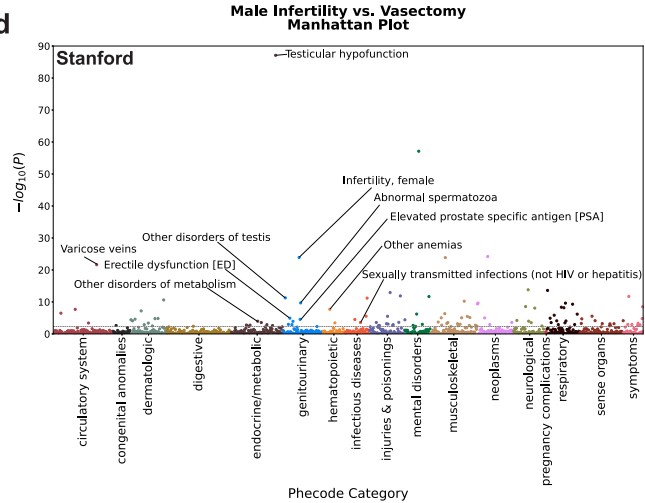

sensitivity analysis, we found 28 positive and 51 negative associations (Supplementary Data 9). We also assessed whether the odds ratios of diagnoses tested for association with MI correlated between the primary analysis and each sensitivity analysis for UC and Stanford and found that the primary analysis significantly correlated with each

sensitivity analysis for both hospital systems (Supplementary Data 10, Supplementary Fig. 8).

We then identified which significant diagnoses were shared across the 3 analyses for UC and Stanford. We found 76 positively (Fig. 4a) and 32 negatively (Supplementary Fig. 9a) associated diagnoses that were

**Fig. 3 | Logistic regression analyses reveal expected and less expected diagnoses associated with male infertility that were first obtained before the 6-month cutoff. a** Volcano plot of patients' diagnosis associations with male infertility for diagnoses first obtained before the 6-month cutoff at UC. **b** Volcano plot of patients' diagnosis associations with male infertility for diagnoses first obtained before the 6-month cutoff at Stanford. **c** Truncated Manhattan plot of diagnoses that were first obtained before the 6-month cutoff at UC, grouped into phecode categories. The following four diagnoses are not included in the plot: Abnormal spermatozoa ($-\log_{10}(p\text{-value}) = 106.74$); Persons encountering health services in circumstances related to reproduction ($-\log_{10}(p\text{-value}) = 231.35$); Testicular hypofunction ($-\log_{10}(p\text{-value}) = 231.35$); and Varicose veins ($-\log_{10}(p\text{-value}) = 115.62$). **d** Manhattan plot of diagnoses that were first obtained before the 6-month cutoff at Stanford, grouped into phecode categories. For panels (**a**) and (**b**), pink dots = $-\log_{10}(p\text{-value})$ of diagnoses significantly associated with male infertility; blue dots = $-\log_{10}(p\text{-value})$ of diagnoses significantly associated with vasectomy-related record; gray dots = $-\log_{10}(p\text{-value})$ of diagnoses that are not significantly associated with male infertility status. Phenotype corresponds to diagnosis. For (**c**), dots indicate the $-\log_{10}(p\text{-value})$ of diagnoses organized by phecode category, where dark red dots—circulatory system, black dots—congenital anomalies, blue-green dots—dermatologic, gold dots—digestive, brown dots—endocrine/metabolic, blue dots—genitourinary, orange dots—hematopoetic, red-orange dots—infectious disease, dark purple dots—injuries & poisonings, dark green dots—mental

disorders, tan dots—musculoskeletal, light purple dots—neoplasms, light green dots—neurological, red dots—null, dark brown dots—pregnancy complications, reddish-brown dots—respiratory, pink dots—sense organs, and medium green dots—symptoms. For panel (**d**), dots indicate the $-\log_{10}(p\text{-value})$ of diagnoses organized by phecode category, where dark red dots—circulatory system, black dots—congenital anomalies, blue-green dots—dermatologic, gold dots—digestive, brown dots—endocrine/metabolic, blue dots—genitourinary, orange dots—hematopoetic, red-orange dots—infectious disease, dark purple dots—injuries & poisonings, dark green dots—mental disorders, tan dots—musculoskeletal, light purple dots—neoplasms, light green dots—neurological, red dots—pregnancy complications, dark brown dots—respiratory, reddish-brown dots—sense organs, and pink dots—symptoms. For panels (**a**, **c**), 1442 diagnoses were compared at UC, and Benjamini-Hochberg correction was used to identify diagnoses significantly associated with male infertility. For panels (**b**, **d**), 1197 diagnoses were compared at Stanford, and Benjamini-Hochberg correction was used to identify diagnoses significantly associated with male infertility. For all panels, top diagnoses found to be significantly associated with male infertility in the primary analysis are annotated. Phecode categories correspond to disease categories. $n = 6531$ UC patients with male infertility; $n = 8353$ UC patients with vasectomy-related record; $n = 5551$ Stanford patients with male infertility, $n = 2464$ Stanford patients with vasectomy-related record. P—$p$-value, UC—University of California.

---

significant in all 3 analyses at UC (Supplementary Data 11). By contrast, we found 19 positively (Fig. 4b) and 49 negatively (Supplementary Fig. 9b) associated diagnoses that were significant across all three analyses at Stanford (Supplementary Data 12). Fifteen significant positively associated diagnoses (Table 2, Fig. 4C, Supplementary Data 13) and 17 significant negatively associated diagnoses (Supplementary Fig. 9c, Supplementary Data 13) overlapped across institutions. These included expected MI comorbidities, such as erectile dysfunction and pituitary hypofunction[23,24]. We also found less expected positively associated comorbidities that were shared across both hospital systems, such as hypothyroidism and other anemias. We also found less expected positively associated comorbidities in the UC analysis specifically. For instance, we found a number of circulatory system conditions, such as circulatory disease, congestive heart failure, and other hypertensive complications in the UC analyses.

### Association analyses of diagnoses first diagnosed beyond 6 months after patients' first male infertility diagnosis or vasectomy-related record reveal circulatory system conditions and other conditions across disease categories at UC

Next, we tested whether any diagnoses that were first obtained beyond 6 months after patients' first MI diagnosis or vasectomy-related record were significantly associated with MI at UC and Stanford. We ran a primary analysis as well as SDoH and hospital utilization sensitivity analyses as described above for the before 6-month cutoff analyses. We also used Benjamini–Hochberg correction to identify significantly associated diagnoses.

We tested 1322 diagnoses in each UC analysis. For the primary analysis, we found 277 positive and 5 negative associations (Fig. 5a, c, Supplementary Data 14). We also tested 1057 diagnoses in each Stanford analysis. For the Stanford primary analysis, we found 72 positive associations (Fig. 5b, d, Supplementary Data 15).

For the UC SDoH sensitivity analysis, we found 271 positive and 2 negative associations (Supplementary Data 14). By contrast, for the UC hospital utilization sensitivity analysis, we found only 21 positive and 13 negative associations (Supplementary Data 14). Similarly, for the Stanford SDoH sensitivity analysis, we found 87 positive associations (Supplementary Data 15). Finally, for the Stanford hospital utilization sensitivity analysis, we found only 3 positive and 21 negative associations (Supplementary Data 15). We also assessed whether the odds ratios of diagnoses tested for association with MI correlated between the primary analysis and each sensitivity analysis for UC and Stanford and found that the primary analysis significantly correlated with each sensitivity analysis for both hospital systems (Supplementary Data 10, Supplementary Fig. 10).

Next, we identified which significant diagnoses were shared across the 3 analyses at UC and Stanford. At UC, while the primary analysis and SDoH sensitivity analysis exclusively shared 240 diagnoses positively associated with MI, only 13 diagnoses were found to be positively associated with MI across all 3 analyses (Fig. 6a, Supplementary Data 16). Moreover, only 2 diagnoses were found to be negatively associated with MI across all 3 analyses at UC (Supplementary Fig. 11, Supplementary Data 16). Additionally, at Stanford, we only found 3 positively associated diagnoses shared across all 3 analyses (Fig. 6b, Supplementary Data 17). There were only two shared comorbidities across all analyses and both hospital systems that were positively associated with MI (Fig. 6c); these were the expected conditions abnormal spermatozoa and testicular hypofunction (Supplementary Data 13). There were also 3 additional expected diagnoses positively associated with MI at UC; these were other disorders of testis, other tests, and persons encountering health services in circumstances related to reproduction (Supplementary Data 16). Interestingly, 8 of the 13 positively associated comorbidities at UC were less expected. These included three circulatory system conditions (arrhythmia, circulatory disease, and other forms of chronic heart disease), other disorders of the kidney and ureters, bacterial enteritis, *H. pylori*, osteoporosis, and other disorders of metabolism (Supplementary Data 16).

We also assessed patients with at least 12, 24, 36, 48, or 60 months of follow-up time beyond 6 months of first being diagnosed with MI or receiving a vasectomy-related record, and we tested for MI-associated comorbidities diagnosed within those timeframes. The number of patients lost for each follow-up cutoff time can be found in Supplementary Data 18 for UC and Stanford. At UC, genitourinary, endocrine/metabolic, and circulatory system diagnoses were enriched for MI patients in the primary analyses. This is most apparent for patients with at least 24 or 36 months of follow-up (Supplementary Data 19). For Stanford, only testicular hypofunction was significantly associated with MI for all fixed follow-up time analyses, with the exception of the 60-month fixed follow-up analyses, in which no significantly associated diagnoses were found (Supplementary Data 20).

### Cox proportional hazards models revealed that male infertility patients have a higher risk of receiving diagnoses found to be significantly associated with male infertility relative to patients with a vasectomy-related record

We then assessed whether MI patients had a higher risk of receiving the 13 diagnoses found to be positively associated with MI across all 3 logistic regression analyses at UC after the 6-month cutoff relative to patients with a vasectomy-related record using Cox proportional hazards models (Fig. 7).

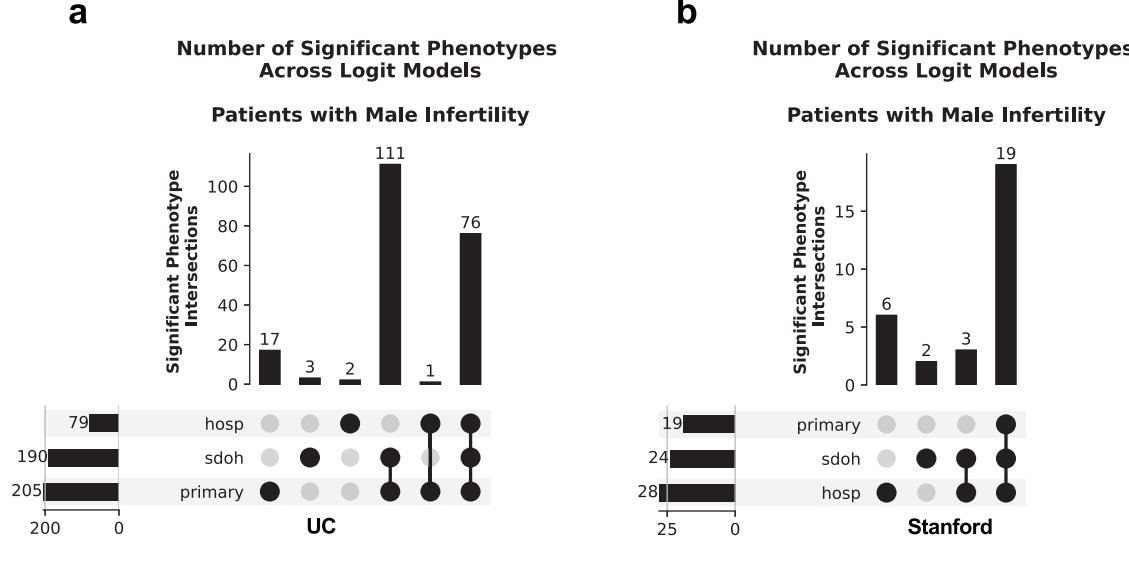

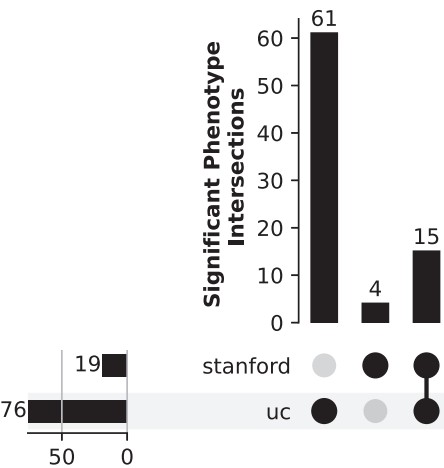

**Fig. 4 | A number of diagnoses are positively associated with male infertility across UC and Stanford before the 6-month cutoff. a** Upset plot of shared significant diagnoses positively associated with male infertility across the primary analysis and both sensitivity analyses for diagnoses first obtained before the 6-month cutoff at UC. **b** Upset plot of shared significant diagnoses positively associated with male infertility across the primary analysis and both sensitivity analyses for diagnoses first obtained before the 6-month cutoff at Stanford. **c** Upset plot of shared significant diagnoses positively associated with male infertility across all six analyses for diagnoses first obtained before the 6-month cutoff across UC and Stanford. For all panels, horizontal rows indicate a specific analysis, and bar charts indicate the number of overlapping significant diagnoses for a given combination of analyses. Phenotypes correspond to diagnoses. hosp—hospital utilization sensitivity analysis, primary—primary analysis, sdoh—social determinants of health sensitivity analysis, uc—University of California. *n* = 6531 UC patients with male infertility, *n* = 8353 UC patients with vasectomy-related record, *n* = 5551 Stanford patients with male infertility, *n* = 2464 Stanford patients with vasectomy-related record.

We found that MI patients had a higher risk of receiving 11 of these diagnoses across all 3 Cox proportional hazards models performed for each diagnosis relative to patients with a vasectomy-related record (Table 3, Supplementary Fig. 12, Supplementary Data 21). The only diagnoses that were not significant across all 3 Cox proportional hazards models were other tests and persons encountering health services in circumstances related to reproduction.

**Comparing primary analyses of diagnoses first obtained prior to versus beyond 6 months after patients' first male infertility diagnosis or vasectomy-related record revealed that disease category representation differs before and after the 6-month cutoff**

We also compared logistic regression primary analysis results for diagnoses first diagnosed before versus after the 6-month cutoff for UC and Stanford.

**Table 2 | Diagnoses positively associated with male infertility prior to six months after receiving a male infertility diagnosis across all logistic regression analyses and both hospital systems**

| phenotype | UC odds ratio | UC CI lower bound | UC CI upper bound | UC p-value | Stanford odds ratio | Stanford CI lower bound | Stanford CI upper bound | Stanford p-value |
|---|---|---|---|---|---|---|---|---|
| Abnormal spermatozoa | 23.61 | 17.81 | 31.28 | 6.52E−105 | 40.44 | 12.94 | 126.47 | 1.42E−08 |
| Anterior pituitary disorders | 7.92 | 3.49 | 17.98 | 1.62E−05 | 10.20 | 2.37 | 43.82 | 2.08E−02 |
| Cancer of prostate | 3.70 | 2.15 | 6.37 | 4.54E−05 | 5.48 | 1.97 | 15.30 | 1.52E−02 |
| Decreased libido | 2.13 | 1.70 | 2.67 | 4.46E−09 | 2.00 | 1.29 | 3.10 | 2.34E−02 |
| Elevated prostate specific antigen [PSA] | 1.97 | 1.34 | 2.90 | 5.15E−03 | 3.43 | 1.93 | 6.09 | 6.49E−04 |
| Erectile dysfunction [ED] | 2.28 | 2.02 | 2.58 | 1.30E−37 | 1.54 | 1.27 | 1.87 | 3.22E−04 |
| Hyperplasia of prostate | 2.08 | 1.64 | 2.63 | 5.48E−08 | 1.76 | 1.26 | 2.47 | 1.38E−02 |
| Hypothyroidism NOS | 1.97 | 1.55 | 2.52 | 1.15E−06 | 1.65 | 1.22 | 2.23 | 1.58E−02 |
| Infertility, female | 48.12 | 11.69 | 197.95 | 2.20E−06 | 26.67 | 14.22 | 50.00 | 3.31E−22 |
| Other anemias | 1.83 | 1.49 | 2.25 | 3.34E−07 | 2.45 | 1.79 | 3.35 | 8.43E−07 |
| Other disorders of testis | 9.22 | 5.09 | 16.68 | 1.83E−11 | 9.47 | 5.00 | 17.94 | 4.85E−10 |
| Pituitary hypofunction | 5.90 | 3.10 | 11.25 | 1.99E−06 | 25.80 | 3.41 | 195.39 | 1.98E−02 |
| Testicular hypofunction | 10.99 | 9.51 | 12.69 | 3.23E−229 | 13.84 | 10.68 | 17.94 | 1.14E + 00 |
| Type 2 diabetes | 2.31 | 1.87 | 2.85 | 9.71E−13 | 1.76 | 1.30 | 2.38 | 4.69E−03 |
| Varicose veins | 5.20 | 4.52 | 5.99 | 1.15E−113 | 5.93 | 4.14 | 8.49 | 4.73E−20 |

Phenotype corresponds to diagnosis.
*UC* University of California.

For UC, we found that the odds ratios of diagnoses significantly associated with MI before and/or after the 6-month cutoff were correlated (Fig. 8a) (Pearson's correlation coefficient = 0.60, *p*-value = 9.31e−42). We found 115 diagnoses significantly associated with MI in both analyses, 85 diagnoses significantly associated with MI only before the 6-month cutoff, and 160 diagnoses significantly associated with MI only after the 6-month cutoff (Supplementary Data 22). For diagnoses that were significant in both analyses, the top three disease categories were endocrine/metabolic conditions (*n* = 30 diagnoses total), such as immunity deficiency and type 2 diabetes, genitourinary conditions (*n* = 20), such as chronic kidney disease stage III and stage IV, and circulatory system conditions (*n* = 16), such as circulatory disease and congestive heart failure (Supplementary Data 22). For diagnoses that were only significant before the 6-month cutoff, the top three disease categories were neoplasms (*n* = 16), such as Hodgkin's disease and leukemia, endocrine/metabolic conditions (*n* = 11), such as disorders of the pituitary gland and its hypothalamic control, and genitourinary conditions (*n* = 9), such as urinary incontinence and renal failure (Supplementary Data 22). For diagnoses that were only significant after the 6-month cutoff, the top three disease categories were circulatory system conditions (*n* = 29), such as pulmonary heart disease and cardiac arrest, digestive conditions (*n* = 15), such as dyspepsia and other specified disorders of function of stomach as well as irritable bowel syndrome, and endocrine/metabolic conditions (*n* = 14), such as other disorders of metabolism and hyperlipidemia (Supplementary Data 22). Overall, for UC, endocrine/metabolic conditions were among the most highly represented regardless of cutoff time. Meanwhile, genitourinary conditions were more highly represented before the 6-month cutoff date as well as regardless of cutoff time, whereas circulatory system conditions were more highly represented after the 6-month cutoff date as well as regardless of cutoff time.

For Stanford, we also found that the odds ratios of diagnoses significantly associated before and/or after the 6-month cutoff were correlated (Fig. 8b) (Pearson's correlation coefficient = 0.69, *p*-value = 8.86e−25). We found 10 diagnoses positively associated with MI in both analyses, 7 diagnoses significantly associated with MI only before the 6-month cutoff, and 62 diagnoses significantly associated with MI only after the 6-month cutoff (Supplementary Data 23). For diagnoses that were significant in both analyses, diagnosis categories included genitourinary conditions (*n* = 4),

including other disorders of testis and hyperplasia of prostate and endocrine/metabolic conditions (*n* = 3), including hypothyroidism and type 2 diabetes (Supplementary Data 23). We also found that varicose veins, other anemias, and other ill-defined and unknown causes of morbidity and mortality overlapped in both analyses (Supplementary Data 23). The 7 diagnoses found to be significant only before the 6-month cutoff were decreased libido, erectile dysfunction, cancer of prostate, other disorders of metabolism, sexually transmitted infections (not HIV or hepatitis), pituitary hypofunction, and neutropenia (Supplementary Data 23). Finally, for diagnoses that were only significant after the 6-month cutoff, the top disease categories were respiratory conditions (*n* = 11), such as allergic rhinitis and asthma, circulatory system conditions (*n* = 8), such as essential hypertension and hypertensive chronic kidney disease, endocrine/metabolic conditions (*n* = 8), such as hyperlipidemia and type 2 diabetes with renal manifestations, and symptoms (*n* = 8), such as abdominal pain and malaise and fatigue (Supplementary Data 23). Overall, Stanford findings share a few broad consistencies with UC findings. Like UC, endocrine/metabolic conditions were among the most highly represented regardless of cutoff time, whereas circulatory system diagnoses were more represented after the 6-month cutoff date.

### Odds ratios of overlapping diagnoses between UC and Stanford correlate

Finally, we assessed whether the odds ratios of diagnoses present in both hospital systems correlated between the UC and Stanford primary analyses for diagnoses first obtained prior to or beyond 6 months after patients' first recorded MI diagnosis or vasectomy-related record. We found that the odds ratios of diagnoses significantly associated with MI at UC and/or Stanford were correlated both before (Fig. 8c) (Pearson correlation coefficient = 0.68, *p*-value = 1.09e−43) and after (Fig. 8d) (Pearson correlation coefficient = 0.24, *p*-value = 9.18e−5) the 6-month cutoff.

### Discussion

In our study, we found both expected and less expected comorbidities that were positively associated with MI at UC and Stanford.

First, we explored whether patients' overall diagnosis profiles differed based on MI status. We also explored whether they differed based on

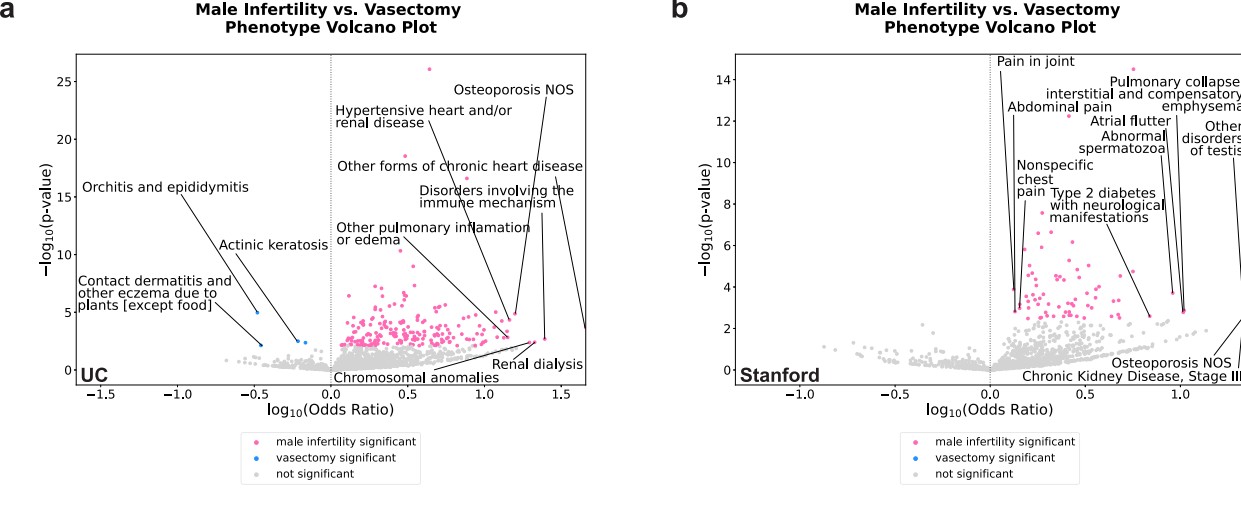

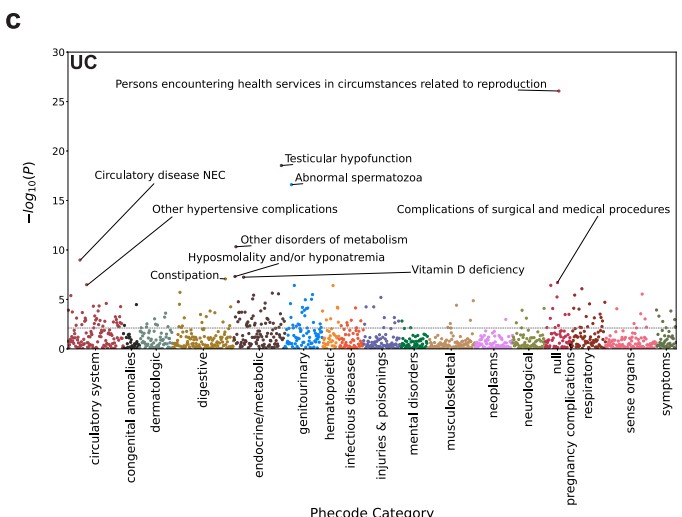

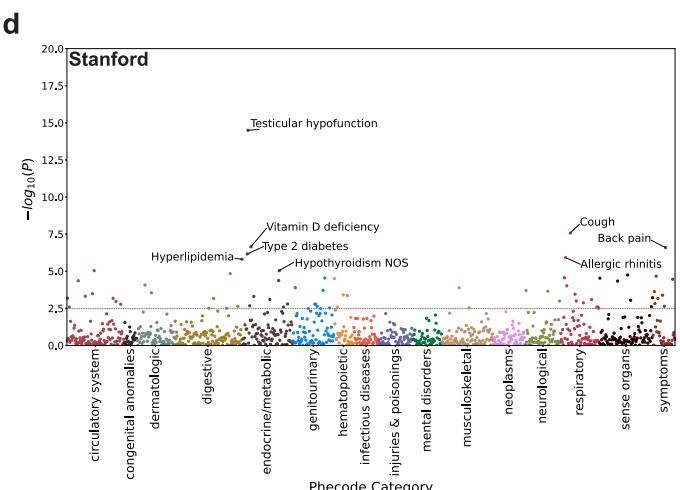

demographic features, location of care, and hospital utilization to determine whether to account for these covariates in our analyses. We found significant differences in patients' diagnoses profiles based on MI status, age, self- or provider-identified race, self- or provider-identified ethnicity, number of visits, and months in the EHR across both hospital systems. Interestingly,

there were fewer significant differences between patients' diagnoses profiles based on MI status for diagnoses first obtained before or after the 6-month cutoff at Stanford relative to UC. For Stanford, only the first UMAP component significantly differed for patients' diagnosis profiles for conditions first obtained before the 6-month cutoff, while no significant differences

**Fig. 5 | Logistic regression analyses reveal expected diagnoses associated with male infertility that were first obtained after the 6-month cutoff across UC and Stanford.** **a** Volcano plot of patients' diagnosis associations with male infertility for diagnoses first obtained after the 6-month cutoff at UC. **b** Volcano plot of patients' diagnosis associations with male infertility for diagnoses first obtained after the 6-month cutoff at Stanford. **c** Manhattan plot of diagnoses that were first obtained after the 6-month cutoff at UC, grouped into phecode categories. **d** Manhattan plot of diagnoses that were first obtained after the 6-month cutoff at Stanford, grouped into phecode categories. For panels (**a**, **b**), pink dots = $-\log_{10}(p$-value$)$ of diagnoses significantly associated with male infertility, blue dots = $-\log_{10}(p$-value$)$ of diagnoses significantly associated with vasectomy-related record; gray dots = $-\log_{10}(p$-value$)$ of diagnoses that are not significantly associated with male infertility status. Phenotype corresponds to diagnosis. For panel (**c**), dots indicate the $-\log_{10}(p$-value$)$ of diagnoses organized by phecode category, where dark red dots—circulatory system, black dots—congenital anomalies, blue-green dots—dermatologic, gold dots—digestive, brown dots—endocrine/metabolic, blue dots—genitourinary, orange dots—hematopoetic, red-orange dots—infectious disease, dark purple dots—injuries & poisonings, dark green dots—mental disorders, tan dots—musculoskeletal, light purple dots—neoplasms, light green dots—neurological, red dots—null, dark

brown dots—pregnancy complications, reddish-brown dots—respiratory, pink dots—sense organs, and medium green dots—symptoms. For panel (**d**), dots indicate the $-\log_{10}(p$-value$)$ of diagnoses organized by phecode category, where dark red dots—circulatory system, black dots—congenital anomalies, blue-green dots—dermatologic, gold dots—digestive, brown dots—endocrine/metabolic, blue dots—genitourinary, orange dots—hematopoetic, red-orange dots—infectious disease, dark purple dots—injuries & poisonings, dark green dots—mental disorders, tan dots—musculoskeletal, light purple dots—neoplasms, light green dots—neurological, red dots—respiratory, dark brown dots—sense organs, and reddish-brown dots—symptoms. For panels (**a**, **c**), 1322 diagnoses were compared at UC, and Benjamini–Hochberg correction was used to identify diagnoses significantly associated with male infertility. For panels (**b**, **d**), 1057 diagnoses were compared at Stanford, and Benjamini–Hochberg correction was used to identify diagnoses significantly associated with male infertility. For all panels, top diagnoses found to be significantly associated with male infertility are annotated. Phecode categories correspond to disease categories. P—p-value, UC—University of California. $n = 6531$ UC patients with male infertility, $n = 8353$ UC patients with vasectomy-related record, $n = 5551$ Stanford patients with male infertility, $n = 2464$ Stanford patients with vasectomy-related record.

were observed in patients' diagnosis profiles for conditions first obtained after the 6-month cutoff.

We found 76 diagnoses at UC and 19 diagnoses at Stanford that were positively associated with MI before the 6-month cutoff, including 15 diagnoses that overlapped across both hospital systems. That we found relatively fewer diagnoses significantly associated with MI at Stanford is consistent with our low-dimensional embedding visualizations, which revealed fewer significant differences in patients' diagnosis profiles based on MI status at Stanford.

A number of comorbidities found to be significantly associated with MI before the 6-month cutoff at UC and Stanford were expected, such as abnormal spermatozoa, testicular hypofunction, and varicose veins. However, we also found a number of less expected comorbidities, such as anemias, hypothyroidism, and prostate cancer.

Prior studies have explored the relationship between male infertility with each of these less expected comorbidities to varying extents. The research on anemia generally is limited, although it is known that sickle cell disease, which can lead to primary and secondary testicular failure, can infer a higher risk for male infertility[25–27]. Importantly, the current report did not identify sickle cell disease as the specific etiology. For hypothyroidism, prior research has been primarily limited to rodent studies or human studies with a small number of participants[28–35]. While thyroid dysfunction is a common cause of female infertility, the evaluation of the infertile male does not include such screening[36]. Finally, studies exploring the relationship between prostate cancer and male infertility often measure the risk of developing prostate cancer after receiving a male infertility diagnosis[37–41]. This is distinct from our finding, which suggests that patients with prostate cancer are receiving this diagnosis prior to or concomitant with their male infertility diagnosis, suggesting that it is possible that these patients are seeking treatment for their infertility, which could be the result of cancer treatment[42]. However, to assess whether the association between prostate cancer and the development of MI could indeed be due to cancer treatment, confounders such as surgery, radiation, and androgen deprivation would need to be controlled for and hopefully will be explored in future studies. The average age of prostate cancer diagnosis occurs in the seventh decade of life, an age beyond which reproduction is usually desired or common. Nevertheless, the age of paternity is increasing, and other groups have reported that reproduction should be increasingly considered among prostate cancer patients[43,44].

We also found several circulatory system diagnoses that were positively associated with MI before the 6-month cutoff in the UC analysis, including circulatory disease, congestive heart failure, and hypertensive complications, with one study suggesting an association between hypertension and impaired semen quality[45].

Our results suggest that it may be beneficial to perform larger, longitudinal studies to clarify the relationship between MI and these comorbidities, which are all treatable conditions. Treatment for hypothyroidism and anemia in particular may be promising for restoring male fertility.

Only 13 and 3 diagnoses were positively associated with MI after the 6-month cutoff (with no fixed follow-up time) in the UC and Stanford analyses, respectively. Abnormal spermatozoa and testicular hypofunction were the 2 diagnoses that overlapped between these hospital systems and were expected.

In the UC analysis, we found that arrhythmia, circulatory disease, and other forms of chronic heart disease were positively associated with MI after the 6-month cutoff. That we found circulatory system conditions is consistent with several longitudinal and cross-sectional studies that suggest an increased incidence of or association with circulatory system conditions for patients with MI[7,11,46–48]. Our study is consistent with these findings and supports testing the hypothesis that MI may be a marker of an elevated risk of developing circulatory system diseases.

Other diseases of the kidney and ureters were also positively associated with MI after the 6-month cutoff in the UC analysis. To our knowledge, it is unknown whether MI leads to an increased incidence of renal-related conditions. One possible interpretation of our finding is that these patients may have had subclinical renal conditions that further developed sometime after their MI diagnosis. Future studies can be done to ascertain whether there is an increased incidence of renal conditions for patients with MI. Additionally, future studies can measure lab values that indicate kidney function before and after a MI diagnosis in patients who then develop a renal-related condition to ascertain whether kidney function may have been compromised before being diagnosed with MI.

Other potentially interesting diagnoses positively associated with MI in the UC analysis after the 6-month cutoff include *H. pylori* infection and osteoporosis; osteoporosis was also significantly associated with MI before the 6-month cutoff in UC. The few studies investigating the relationship of MI with these and other related conditions are inconclusive and conflicting[49–55]. Our findings indicate that larger studies clarifying the relationship between MI and these conditions may be beneficial.

Notably, Cox proportional hazards models revealed that MI patients had a higher risk of receiving 11 of the 13 diagnoses found to be significantly associated with MI after the 6-month cutoff at UC relative to patients with a vasectomy-related record. This demonstrates that we can use findings from the logistic regression association analyses in longitudinal studies to further clarify the relationship between MI and associated comorbidities.

When we compared the effect sizes of diagnoses from the primary analysis before and after the 6-month cutoff, we found that they were

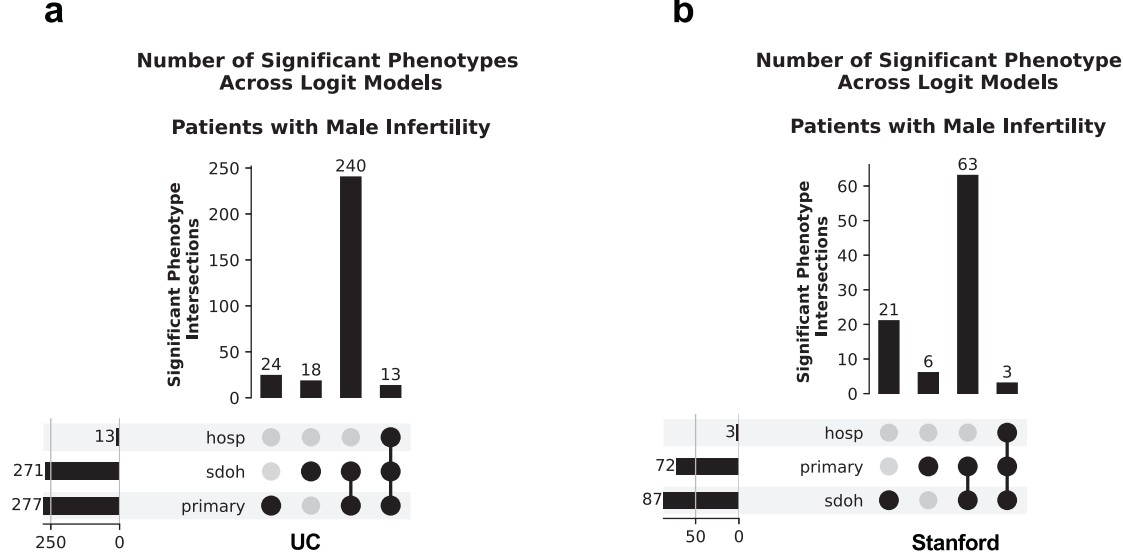

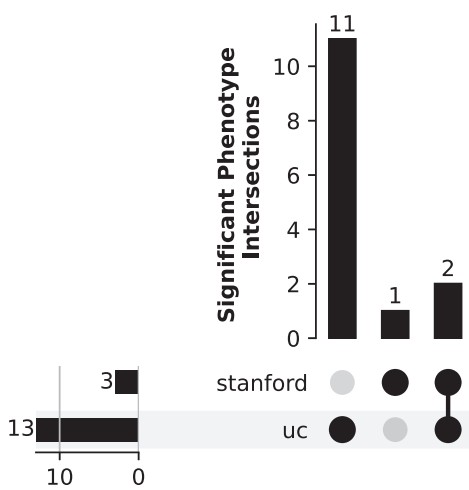

**Fig. 6 | A number of diagnoses are positively associated with male infertility across UC and Stanford after the 6-month cutoff. a** Upset plot of shared significant diagnoses positively associated with male infertility across the primary analysis and both sensitivity analyses for diagnoses first obtained after the 6-month cutoff at UC. **b** Upset plot of shared significant diagnoses positively associated with male infertility across the primary analysis and both sensitivity analyses for diagnoses first obtained after the 6-month cutoff at Stanford. **c** Upset plot of shared significant diagnoses positively associated with male infertility across all six analyses for diagnoses first obtained after the 6-month cutoff across UC and Stanford. For all panels, horizontal rows indicate a specific analysis, and bar charts indicate the number of overlapping significant diagnoses for a given combination of analyses. Phenotypes correspond to diagnoses. hosp—hospital utilization sensitivity analysis, primary—primary analysis; sdoh—social determinants of health sensitivity analysis, uc—University of California. $n = 6531$ UC patients with male infertility, $n = 8353$ UC patients with vasectomy-related record, $n = 5551$ Stanford patients with male infertility, $n = 2464$ Stanford patients with vasectomy-related record.

significantly correlated in both hospital systems, suggesting that there are a number of conditions that may be associated with MI regardless of cutoff time. Interestingly, however, disease category representation may depend on when they were first diagnosed relative to MI. Genitourinary diseases, including renal conditions, tend to be more associated with MI before the 6-month cutoff, while circulatory system diseases tend to be more associated with MI after the 6-month cutoff. Moreover, endocrine/metabolic

conditions seem to be associated with MI regardless of cutoff time. This suggests that it is possible that different disease categories may be more likely to be risk factors for or adverse health outcomes of MI. Future hypothesis-driven, longitudinal studies utilizing EHR could clarify the relationship between the manifestation of certain conditions and the development of MI. For example, a retrospective cohort study can test whether patients with renal conditions are more likely to develop MI.

**Fig. 7 | Overview of time-to-event measurements for Cox proportional hazards models.** How time-to-event was measured for patients with and without the diagnosis being tested.

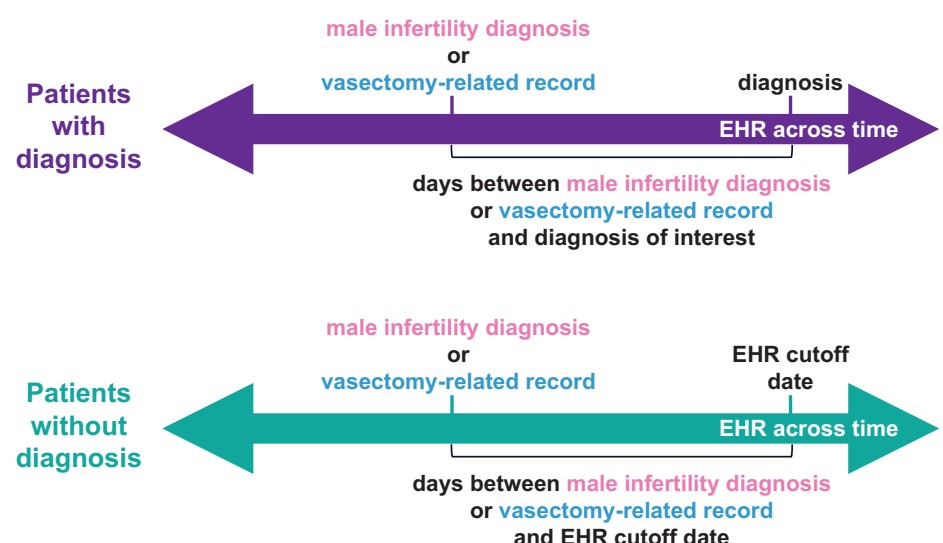

**Table 3 | Results from Cox proportional hazards models for the 13 diagnoses found to be significantly associated with male infertility after the 6-month cutoff at UC**

| phenotype | hazard ratio | CI lower bound | CI upper bound | standard error | statistic | *p*-value |
|---|---|---|---|---|---|---|
| Abnormal spermatozoa | 17.86 | 12.72 | 25.06 | 0.17 | 16.66 | 2.39E−62 |
| Arrhythmia (cardiac) NOS | 2.51 | 1.51 | 4.17 | 0.26 | 3.53 | 4.09E−04 |
| Bacterial enteritis | 2.82 | 1.50 | 5.30 | 0.32 | 3.23 | 1.23E−03 |
| Circulatory disease NEC | 3.16 | 2.31 | 4.33 | 0.16 | 7.18 | 7.23E−13 |
| H. pylori | 2.50 | 1.50 | 4.17 | 0.26 | 3.53 | 4.22E−04 |
| Osteoporosis NOS | 8.55 | 3.27 | 22.32 | 0.49 | 4.38 | 1.19E−05 |
| Other disorders of metabolism | 2.28 | 1.74 | 2.99 | 0.14 | 5.94 | 2.89E−09 |
| Other disorders of testis | 17.14 | 6.10 | 48.18 | 0.53 | 5.39 | 7.13E−08 |
| Other disorders of the kidney and ureters | 2.28 | 1.61 | 3.24 | 0.18 | 4.62 | 3.76E−06 |
| Other forms of chronic heart disease | 10.77 | 3.77 | 30.78 | 0.54 | 4.44 | 9.06E−06 |
| Other tests | 1.20 | 1.11 | 1.31 | 4.37E-02 | 4.26 | 2.02E−05 |
| Persons encountering health services in circumstances related to reproduction | 0.62 | 0.55 | 0.70 | 6.22E-02 | −7.74 | 9.71E−15 |
| Testicular hypofunction | 8.63 | 7.26 | 10.27 | 8.85E-02 | 24.36 | 4.51E−131 |

Results are adjusted for age at first male infertility diagnosis or vasectomy-related record and UC location (primary analysis; see *Cox Proportional Hazards Model* in Methods for details).
*CI lower bound* lower bound of 95% confidence interval for hazard ratio associated with having male infertility, *CI upper bound* upper bound of 95% confidence interval for hazard ratio associated with having male infertility.

We also compared the effect sizes of diagnoses significantly associated with MI in the primary analysis at UC and/or Stanford. We found that they were significantly correlated before and after the 6-month cutoff, with the correlation being higher before the 6-month cutoff. That we found a higher correlation before the 6-month cutoff may be consistent with our temporally-stratified association analysis findings and low-dimensional embedding visualizations, where we found more diagnoses significantly associated with MI at Stanford before the 6-month cutoff as well as more differences in patients' diagnosis profiles based on MI status before the 6-month cutoff.

There are several limitations to consider in our study. First, the logistic regression study design and its lack of accounting for time limits the interpretation of findings, in particular the directionality of associations. Thus, longitudinal analyses are required to assess directionality. To address this, we used Cox proportional hazards models to further assess the relationship between male infertility and the 13 diagnoses found to be associated with male infertility for UC patients after the 6-month cutoff. Second, some of the differences observed in the UMAP of diagnosis profiles between MI patients (cases) and vasectomy patients (controls) might be due to different

screening practices for MI diagnoses and vasectomy procedures. Evaluations for both MI and vasectomy, however, mandate a thorough acquisition of medical history as well as a physical examination to evaluate for comorbidities. While some conditions may be related to a specific screening practice, such as identifying varicoceles during an MI evaluation, many others, such as hypothyroidism, would be unlikely to be diagnosed during an MI evaluation. Third, there could be differences between MI patients and vasectomy patients that are causally associated with the observed outcomes but are not due to MI. While our sensitivity analyses try to account for this, they likely do not account for all potential bias, such as potential differences in the health-seeking behaviors of MI patients and vasectomy patients. Moreover, there is inherent uncertainty about when patients first developed MI; we only know when patients were first diagnosed with the condition at UC or Stanford. Generally, we cannot know for certain when patients first developed any particular medical condition using EHR data. We are also unable to tell whether they were first diagnosed outside of UC or Stanford due to the legal limitations placed on systematically sharing personal health information across institutions; this potential missingness is an inherent limitation of studies utilizing EHR data. As a result, we cannot know for

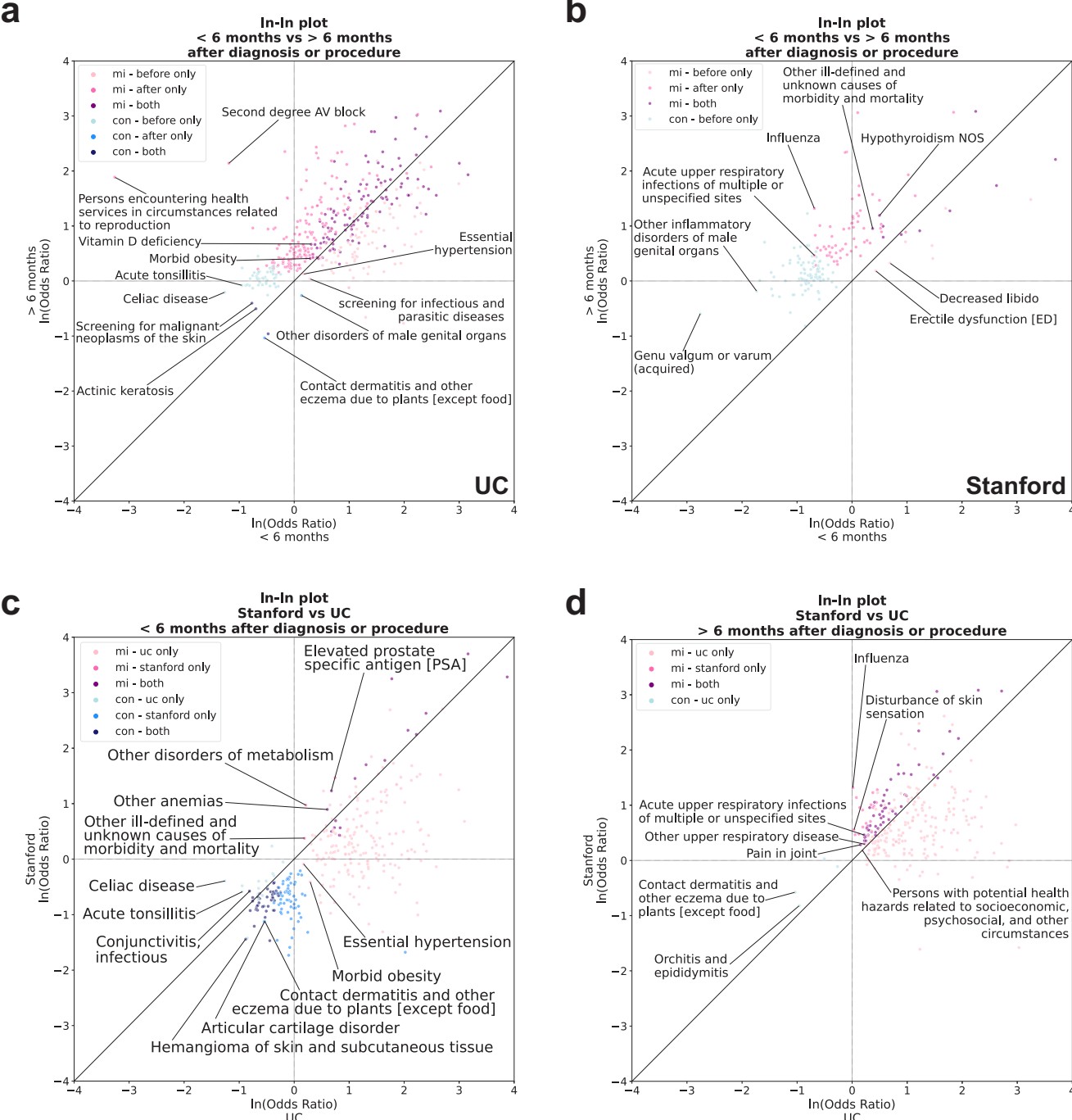

certain whether the comorbidities we identified manifested before, concurrent with, or after the development of MI.

Another limitation is that we did not perform a power analysis; thus, some of our findings may be false negatives. We also did not estimate false positives from correlated diagnoses. Thus, some of our findings may be false positives, which we attempted to mitigate by identifying significant male infertility associated diagnoses not only in the primary analysis, but also in the SDoH and hospital utilization sensitivity analyses across both UC and Stanford.

Additionally, we selected patients with any vasectomy record, including those who underwent vasectomy reversal or who had follow-up tests after their vasectomy. Thus, for some vasectomy patients, some diagnoses before the 6-month cutoff may have occurred 6 months after their

vasectomy. Even with this caveat, the vast majority of UC patients in our study (> 98%) have either a known vasectomy date or recorded post-vasectomy semen analysis, which usually occurs around 3 months after a vasectomy[56]. Moreover, vasectomy patients were chosen for their presumptive fertility, as previous studies have shown that 90% of patients seeking a vasectomy are fertile[12]. Thus, that the precise date of vasectomy may be unknown may not have a meaningful impact on our findings.

Many UC associations were not significant in the Stanford analysis, which was particularly the case after the 6-month cutoff. This could be due to several reasons. First, there were far fewer vasectomy patients in the Stanford analysis, which likely reduced statistical power for the association analyses. Additionally, there may be multiple differences in the overall patient population at UC and Stanford, which may limit how many

**Fig. 8 | Significant diagnoses in the primary analysis are correlated before and after the 6-month cutoff as well as between UC and Stanford. a** Ln-ln plot of the ln(odds ratio) of significant diagnoses in the primary analysis that were first obtained after the 6-month cutoff versus the ln(odds ratio) of significant diagnoses in the primary analysis that were first obtained before the 6-month cutoff at UC. **b** Ln-ln plot of the ln(odds ratio) of significant diagnoses in the primary analysis that were first obtained after the 6-month cutoff versus the ln(odds ratio) of significant diagnoses in the primary analysis that were first obtained before the 6-month cutoff at Stanford. **c** Ln-ln plot of the ln(odds ratio) of significant diagnoses in the primary analysis that were first obtained before the 6-month cutoff at Stanford versus the ln(odds ratio) of significant diagnoses in the primary analysis that were first obtained before the 6-month cutoff at UC. **d** Ln-ln plot of the ln(odds ratio) of significant diagnoses in the primary analysis that were first obtained after the 6-month cutoff at Stanford versus the ln(odds ratio) of significant diagnoses in the primary analysis that were first obtained after the 6-month cutoff at UC. For all panels, Pearson correlation coefficient was used to assess correlation and significance. For panels (**a**, **b**), mi - before only (light pink dots)—diagnoses that are positively associated with male infertility before the 6-month cutoff only, mi - after only (pink dots)—diagnoses that are positively associated with male infertility after the 6-month cutoff only, mi - both (dark pink dots)—diagnoses that are positively associated with male infertility both before and after the 6-month cutoff, con - before only (light blue dots)—diagnoses that are negatively associated with male infertility before the 6-month cutoff only, con - after only (blue dots)—diagnoses that are negatively associated with male infertility after the 6-month cutoff only, con - both (dark blue dots)—diagnoses that are negatively associated with male infertility both before and after the 6-month cutoff. For panels (**c**, **d**), mi - uc only (light pink dots)—diagnoses that are positively associated with male infertility at UC only, mi—stanford only (pink dots)—diagnoses that are positively associated with male infertility at Stanford only, mi - both (dark pink dots)—diagnoses that are positively associated with male infertility at UC and Stanford, con - uc only (light blue dots)—diagnoses that are negatively associated with male infertility at UC only, con - stanford only (blue dots)—diagnoses that are negatively associated with male infertility at Stanford only, con - both (dark blue dots)—diagnoses that are negatively associated with male infertility at UC and Stanford. UC—University of California. $n = 6531$ UC patients with male infertility, $n = 8353$ UC patients with vasectomy-related record; $n = 5551$ Stanford patients with male infertility, $n = 2464$ Stanford patients with vasectomy-related record.

significant findings overlap between the two hospital systems. It is also possible that the extract, transform, and load (ETL) process underlying the transfer of patient records from the hospitals' EHR to de-identified or limited data set databases systematically differ between the two hospital systems, which may lead to differences in diagnosis representation. Nonetheless, it is promising that the effect sizes of overlapping diagnoses were significantly correlated between UC and Stanford, even if the diagnoses themselves were not necessarily significantly associated in both institutions. Future studies can perform association analyses with larger numbers of case and control patients to explore whether similar comorbidities are significantly associated with MI.

Another important consideration is potentially reduced sensitivity for identifying MI patients in the EHR. The specific codes used for patients may vary by provider and the vagaries of insurance coverage. Thus, administrative differences amongst physicians may affect how patients' conditions are represented in the EHR. This is indeed an inherent limitation for any EHR study that uses administrative/insurance claims to study disease[57].

Moreover, the sensitivity and specificity of our criteria for male infertility and vasectomy are inherently incalculable in the UC and Stanford EHR due to a lack of accessible ground truth. That is, we do not know how many patients truly have male infertility or have had a vasectomy. In general, having broad inclusion criteria would maximize sensitivity while potentially compromising specificity, while more stringent inclusion criteria generally will increase specificity while decreasing sensitivity. For example, a 2017 review demonstrated that of the patients who have undergone a semen analysis and have been diagnosed with at least one of five male infertility ICD-9 codes (606.0, 606.1, 606.8, 606.9, and V26.21), 92.4% were determined to have abnormal semen quality[58]. If the patients had at least three distinct infertility codes, this rises to 99.8%[58].

We also found that adjusting for hospital utilization substantially affected the number of significant comorbidities positively associated with MI, particularly for diagnoses first obtained after the 6-month cutoff. This may be due to increased healthcare utilization for patients with MI relative to post-vasectomy patients, leading to more opportunities to be diagnosed with a particular condition. This underscores the importance of accounting for potential differences in healthcare utilization when performing EHR-based studies.

Another limitation is that our patient cohort may not be representative of the overall population experiencing MI. Male infertility counseling is done relatively less often compared to female infertility counseling, which may in part be due to social stigma[59]. This suggests that there may be patients experiencing MI who are not seeking infertility care. Moreover, it is possible that those seeking infertility treatment may be disproportionately more likely to afford expensive fertility treatments relative to the general population. Finally, there are likely individuals who are unaware they have MI,

especially if they are not pursuing family planning. Thus, our study of MI may have limited generalizability.

There have also been studies suggesting that environmental exposures may play a role in MI and declining sperm count[60]. However, environmental exposures are not systematically captured in either of the EHR databases utilized in this study. Future work can ascertain the relationship between MI, comorbidities, and environmental exposures by taking into account where patients live.

Overall, our data-driven approach revealed expected and less expected comorbidities associated with MI across UC and Stanford. Our temporally-stratified association analyses point to conditions that we hypothesize may be risk factors for or adverse outcomes of MI. Our findings can serve as a useful starting point for clarifying the relationship between these conditions and MI in future longitudinal studies. Ultimately, we hope that this study will help uncover new etiologies underlying currently idiopathic MI as well as actionable insights for prevention and treatment.

## Data availability

UCDDP is unavailable to those who are unaffiliated with UC, and Stanford's de-identified EHR is unavailable to those who are unaffiliated with Stanford. UC researchers may access UCDDP after an initial data analysis within their respective UC health center supports a UC-wide extension; details on how to access UCDDP for UC researchers can be found in the University of California Health Center for Data-driven Insights and Innovation website (https://www.ucop.edu/uc-health/departments/center-for-data-driven-insights-and-innovations-cdi2.html). Approved Stanford researchers may access the Stanford de-identified EHR through the Stanford Research Repository (https://med.stanford.edu/starr-tools). Source data used to generate Figs. 3–6 and 8 as well as Supplementary Figs. 8-11 are provided in Supplementary Data files 8, 9, and 13-15. Source data for logistic regression fixed follow-up time analyses are provided in Supplementary Data files 19 and 20. Source data for Cox proportional hazards models are provided in Supplementary Data file 21. Note that the source data in supplementary data files are censored to protect patient privacy; patient counts less than or equal to 10 are set to 10. We suggest obtaining similar numbers of male infertility patients and vasectomy patients in this study for the reproducibility of findings.

## Code availability

The code developed for this study is available at https://doi.org/10.5281/zenodo.16058958[61].

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

## Acknowledgements
Primary support was provided by the March of Dimes (M.S., N.A.) and the UCSF Center for Reproductive Sciences T32HD007263 (S.R.W.). Additional support was provided by the Medical Scientist Training Program T32GM007618 (A.S.T.), National Science Foundation Graduate Research Fellowship Program Grant No. 2038436 (J.R.), R01HD105256 (M.S., A Rajkovic, R.B.L., N.A.), R35GM138353 (N.A.), 1R01HL139844 (N.A.), 3P30AG066515 (N.A.), 1R01AG058417 (N.A.), P01HD106414 (N.A.), the Burroughs Wellcome Fund (N.A.), the Alfred E. Mann Foundation (N.A.), and the Robertson Foundation (N.A.). The authors acknowledge the use of resources developed and supported by the UCSF IT Academic Research Systems and the UCSF Bakar Computational Health Sciences Institute Information Commons teams and thank members of these teams for technical support. The authors also thank the Center for Data-driven Insights and Innovation at UC Health (CDI2; https://www.ucop.edu/uc-health/departments/center-for-data-driven-insights-and-innovations-cdi2.html), for its analytical and technical support related to use of the UC Health Data Warehouse. This research used data or services provided by the STAnford medicine Research data Repository (STARR), a clinical data warehouse containing live EPIC data from Stanford Health Care (SHC), the Stanford Children's Hospital (SCH), the University Healthcare Alliance (UHA) and Packard Children's Health Alliance (PCHA) clinics and other auxiliary data from hospital applications such as radiology PACS. STARR platform is developed and operated by the Stanford Medicine Research IT team and is made possible by the Stanford School of Medicine Research Office. We would like to thank the UCSF/Stanford Trio analysis of Recurrent pregnancy loss Integrated bioinformatics genomics Study (TRIOS) for feedback on the manuscript. We would also like to thank Jean Costello and all members of the Sirota lab for their input and advice.

## Author contributions
S.R.W., A Roldan, A.S.T., M.E., and M.S. formulated the study design. S.R.W., J.R., and A.S.T. formulated the statistical analyses. I.E.A. reviewed the statistical analyses. S.R.W., J.R., A.S.T., and T.T.O. navigated UCDDP data access and mapping ICD diagnoses to phecodes. S.R.W., J.R., F.X., N.A., and M.S. determined how to port the analysis pipeline from UC to Stanford. S.R.W. carried out the UC analyses, and F.X. carried out the Stanford analyses. M.E. provided clinical insights for interpreting the results. S.R.W. and F.X. generated the figures. S.R.W. wrote the manuscript. S.R.W., J.R., A.S.T., T.T.O., D.K.S., R.B.L., A Rajkovic, M.E., and M.S. contributed to editing the manuscript. All authors read and approved the final manuscript.

## Competing interests
The following authors declare the following competing interests: J.R. was a Roche intern. M.E. is an advisor to Doveras and Next. All other authors declare no competing interests.

## Additional information

[1]Bakar Computational Health Sciences Institute, University of California San Francisco, San Francisco, CA, USA. [2]Department of Anesthesiology, Perioperative, and Pain Medicine, Stanford University, Stanford, CA, USA. [3]Department of Pediatrics, Stanford University, Stanford, CA, USA. [4]Department of Biomedical Data Science, Stanford University, Stanford, CA, USA. [5]Division of Computational Health Sciences, Department of Surgery, University of Minnesota, Minneapolis, MN, USA. [6]Division of Clinical Informatics and Digital Transformation, Medicine, University of California, San Francisco, San Francisco, CA, USA. [7]Department of Obstetrics and Gynecology, Stanford University, Stanford, CA, USA. [8]Department of Pathology, University of California San Francisco, San Francisco, CA, USA. [9]Institute of Human Genetics, University of California San Francisco, San Francisco, CA, USA. [10]Department of Obstetrics, Gynecology and Reproductive Sciences, University of California, San Francisco, San Francisco, CA, USA. [11]Department of Epidemiology and Biostatistics, University of California, San Francisco, San Francisco, CA, USA. [12]Department of Urology, Stanford University, Stanford, CA, USA. ✉e-mail: eisenberg@stanford.edu; Marina.Sirota@ucsf.edu

