## [Transparent Peer Review file · Communications Medicine]

Leveraging Electronic Health Records from Two Hospital Systems Identifies Male Infertility Associated Comorbidities Across Time

Corresponding Author: Dr Marina Sirota

Version 0:

Reviewer comments:

Reviewer #1

(Remarks to the Author)

Thank you for the opportunity to review the manuscript entitled “Leveraging Electronic Medical Records from Two Hospital Systems Identifies Male Infertility Associated Comorbidities Across Time”.

As context, I am experienced with EHR data and modeling thereof, but am neither an expert in infertility or in association studies using EHR data.

This manuscript describes a multi-site analysis of diagnoses identified following a male infertility diagnosis. It studies an important, understudied problem. The study is well-designed, the report is very clearly written, and the conclusions seem reasonable.

However, in the context of the journal’s aim to publish findings that “represent significant advances in preventing, diagnosing, or treating human disease” and criteria for publication, it does not appear to me that this manuscript reports sufficient impact and/or innovation. The approach of using multiple sensitivity analyses and multiple systems’ data seems solid, but no claims are made regarding the novelty of the methods or approach. There are interesting associations identified that hold-up to sensitivity, multi-site, and clinical analyses; but, the proposed consequences are further study to better assess significance.

Specific comments:

- How sensitive and specific is the MI diagnosis criteria applied?

- Who are the outliers in the UMAP in Figure 2? Cluster appears associated with EHR utilization? Expect these would affect the embedding space considerably. Why is further refinement of the data not called for?

- As discussed, even for findings that held-up in sensitivity analyses and across institutions, assessing clinical significance is challenging. The authors cite a need for longitudinal analyses. Are there further analyses that could be performed using available EHR data that could provide complementary support for these findings, such as time-to-event analyses, association using validated computable phenotypes for a comorbidity, or analyses indexed on the comorbidity.

Reviewer #2

(Remarks to the Author)

The authors discuss the association between medical conditions and MI using 2 separate EMR databases to determine what conditions are associate w/ MI. Specifically looking at temporal relationships of diagnosis (within 6 months vs after). The outcomes are certainly meriting publication however the manuscript is very difficult to interpret in its current condition. There is a significant amount of data presented to the point it dilutes some of the important findings. For example, the discussing regarding the association of diabetes (which has been published on previously) overshadows the more novel findings of potentially missing different CV diagnoses. I would consider, condensing the results and discussion section to

either focus on these findings OR focus on the amalgamation of diagnoses (expected vs unexpected in aggregate) rather than focus on the specific diagnosis.

Abstract:

The wording is awkward in lines 21 and 28-32, consider revising for clarity.

Line 21 - Consider stating MI "solely" can be attributed to 30% of the cases of infertility or discuss that MI may be implicated in up to 50% (in tandem with female factors) as discussed in the introduction.

Discussion:

Lines 363-379 state that the relationship between DM and MI has "not been well-established" but cite 10 papers discussing the relationship, would consider removing this statement.

Results:

I appreciate the transparency of this section. I would strongly consider moving a portion of the results to the methods and/or even a supplemental methods section to improve the readability. For example, lines 87-94 describe the methodology and then are followed by the results.

Likewise, lines 155-158 summarize the findings from the aforementioned several paragraphs and would be more concise for readers.

Methods:

There needs to be a discussion of what conditions are expected and what conditions are not expected based on the literature. For example DM is considered "not expected" though the relationship has been published on significantly i.e. PMIDs 25814158, 29887834, 26674559.

Version 1:

Reviewer comments:

Reviewer #1

(Remarks to the Author)

Thank you for the opportunity to review the resubmission of manuscript entitled "Leveraging Electronic Medical Records from Two Hospital Systems Identifies Male Infertility Associated Comorbidities Across Time". The revisions have improved readability and interpretability considerably. I expect the approach and findings will be interesting to the readership. I provide the below suggestions to further highlight findings most useful for readership.

Comments:

- All the different analyses provide insight into the quality and interpretability of the study. However, the presentation is very complicated for assessing any individual clinical question. If the goal is to 'set the groundwork for future studies to clarify the relationship between less expected comorbidities and MI', consider presenting a view of the data relative to specific clinical hypotheses rather than across all clinical hypotheses. Consider highlighting some of the less expected comorbidities by pulling together a set of the analyses (e.g. ORs and p-values for several sub-analyses) together into a single view/table. This would help in assessing the significance of diagnoses/diagnosis groups highlighted in the discussion. If this frame is buried in a supplemental table, then I have missed it.

- Please include a limitation referring to the selection of control population. The motivation for selecting this control population is well described. But, there could be other differences between cases and controls that are causally associated with the observed outcomes but are not due to MI. Table 1 shows many differences, notably include race identified as Asian and utilization. Sensitivity analyses appear reasonably designed to try to account for this, but do not account for all potential bias.

- Within 6 months, why does it make sense to observe just as many significant negative associations as significant positive associations given the design and the observation in the post-6 month data? Does this not suggest that many of the positive associations are likely false-positives?

- Improve resolution of supplemental figures.

- Code repos is not accessible

- Please explain why it is not preferable to use apply a common UMAP transformation across analyses to facilitate comparisons, such as between UC and Stanford.

Reviewer #2

(Remarks to the Author)

The updated manuscript is much more readable in its current state and emphasizes potentially novel findings surrounding

anemia, thyroid dysfunction, and male infertility.

I challenge the association between prostate cancer and the development of male factor infertility, being significant only due to iatrogenic factors. Is there a way to control for surgery, radiation, and androgen deprivation in this cohort? If not, I would either remove the association given these significant limitations or state this limitation explicitly.

Reviewer #3

(Remarks to the Author)

The authors present a data driven approach to identify whether Male Infertility (MI) compared to vasectomy (health controls) was associated with non-MI disease diagnoses. They specifically focus on non-MI diagnoses between 0 and 6 months after an MI/VAS diagnosis and, non-MI diagnoses after the 6 months period.

My comments are as follows.

1. The logistic regression analyses use MI/VAS as the outcome variable, this should be the exposure/independent variable. Using MI/VAS as the exposure variable ensure the odds ratio's can be readily interpreted which provide information on the magnitude of association.
2. In line with the previous comment please report the odds ratio in a supplementary table (for the primary and sensitivity analyses). Why do the figure plot the $\log_{10}(\text{odds ratio})$, the natural logarithm of the odds ratio (the regression coefficients of a logistic regression model) is more sensible.
3. While I understand the desire to look at short/medium and long-term associations, the nature of the EHR data make it highly likely that diagnoses within the 0 to 6 months period were already present before the MI diagnosis. This of course very much depends on the health seeking behaviour of a participant, which clearly will be distinct for MI patients (these will be more thoroughly checked than VAS). As such differences in non-MI disease diagnoses are to be expected, especially within this short/medium term follow-up period. Taking a more biological approach how would the authors expect MI would lead to de novo disease development such as the onset of heart failure (figure 3.a)? This is not to say that the analysis is wrong, it should just be extremely clear that this analysis is likely picking up pre-existing conditions.
4. The analysis looking at outcomes after 6 months is difficult to interpret due to the open ended follow-up period. Did the outcome occur 7 months after MI diagnosis or 7 years? Possibly the analysis can be enriched by providing the median and 1st and 3rd quartiles of the diagnosis time relative to the VAS/MI diagnosis?
5. In fact a logistic regression model (which uses a Binomial distribution) is invalid when the follow-up time is not fixed. So please fix the follow-up time to something sensible (say 5/10 years).
6. In line with this follow-up time comment, after settling on a fixed time period, please indicate how many participants were lost to follow-up?
7. Depending on the amount of censoring more formal survival methods such as Cox's regression model might be relevant. Such a model would be able to natively account for censoring.
8. Please clarify what would have happened if a participant got a diagnosis in a different hospital. Would this diagnosis be registered in the currently used data? If not, how can the researcher be sure about the exposure and outcome status?
9. While I appreciate the UMAP analysis, is it really that surprising that people with MI have a different diagnosis profile when compared to people with VAS? MI people get screened while VAS do not get screened – this alone would ensure there is a difference in diagnosis.

Overall the figure are nicely formatted.

Figure 3.c uses a piece wise y-axis, this may well be erroneously interpreted. Please use a normal continuous y-axis, if needed one could consider truncating the y values above a certain threshold. This truncation should of course be explained and the un-truncated values presented someplace (e.g. in the figure note if there are not too many)

Figure 3.d, Figure 4 c-d need y-axis labels.

Figure 7, the resolution is too low.

Version 2:

Reviewer comments:

Reviewer #1

(Remarks to the Author)

Thank you for the opportunity to review this resubmission of manuscript entitled "Leveraging Electronic Medical Records from Two Hospital Systems Identifies Male Infertility Associated Comorbidities Across Time". The authors have reasonably addressed all of my previous comments.

Reviewer #3

(Remarks to the Author)

I am grateful to the authors for their detailed response. The authors now indicate that any diagnoses made in a different

hospital were not recorded, as such the exposure and outcome variables are incorrect, invalidating their analysis and the present manuscript. The authors remark that EHR data always suffers from this type of incomplete record keeping is of course incorrect, countries like Estonia or the UK have national coverage severely limiting this issue. The current problem is so fundamental that even if the comments below could be addressed the study is still faulty beyond repair.

Furthermore, while the authors employ a binomial regression model which fundamentally assumes equal follow-up time, with anybody with a shorter follow-up time (and without an event) should be treated as having missing data, the authors choose to ignore this and simply assign these people to being free of disease. This is completely incorrect.

If the authors want to use the full follow-up either using imputation to deal with people that have shorter follow-up and apply a binomial model, or use a time-to-event approach that allows for censoring.

In part due to the above issue, the authors use the outcome as the dependent variable and the exposure as an independent variable. As indicated before this is incorrect, and prohibits straightforward interpretation of the results.

The authors reply stating that the diagnostic work-up for people with MI and male infertility is equal is clearly incorrect. It seems highly unlikely that people with VAS for example routinely receive an ECG or even the biochemistry would be distinct. As such even when there would be complete data, including information from other hospitals or GPs, such a comparison is biased due to differences in diagnostic procedure.

Version 3:

Reviewer comments:

Reviewer #1

(Remarks to the Author)

Reviewer #3

(Remarks to the Author)

I am grateful to the authors for raising this appeal, clearly I was mistaken in the comparison (fully my fault for paying insufficient time on this). Their letter has helped with this.

Two things I would like to raise, one suggestion and one remaining confusion.

The suggestion: it seems the authors design might best be described as a nested case-control study, nested within a cohort of two hospital and where exposure variables are collected through a mixture of hospital records and self-reported data. It would help a lot if the authors could describe the study a long these lines and explain where and when the study deviated from this design.

The part I am still confused about. The paper conducts two analysis, a retrospective analysis and a prospective analysis. In the retrospective analysis the authors indeed identify risk factors for male infertility. In the prospective analysis the authors say they are trying to "identify comorbidities that may have manifested after a MI diagnosis and thus could be adverse outcomes of MI". This implies that MI is an exposure and the diagnosis after MI are considered outcomes – which are subject to missing data due to shorter follow-up or because some diagnoses occurred in a different healthcare settings. Why do the authors still use MI as the outcome variable in their logistic regression analysis? A time to event analysis would at least deal with people having less follow-up (i.e., people with 2 and 4 year follow-up could be combined, rather than setting the person with 2 years follow-up at missing)? Furthermore for the prospective analysis there seems to be a much larger problem with outcomes not being registered in the two hospital. I do agree that for the retrospective analysis this is much less of a concern and the employed analysis makes sense.

Version 4:

Reviewer comments:

Reviewer #3

(Remarks to the Author)

No further comments.

Reviewer #4

(Remarks to the Author)

As raised by another reviewer, it is not entirely clear how time is taken into account in the design and analysis. Ideally it would seem that a nested case-control study taking time into account could have been designed, which would be followed by a time-to-event analysis. It seems that the authors have considered a nested case-control-type design which disregards time, and then (appropriately) used logistic regression for the analysis. It may be though that because of the structure of the underlying data patients with and without male infertility contribute different amounts of time. Has this been taken into account?

Reviewer #5

(Remarks to the Author)

In my view the authors have addressed most of the prior reviewer concerns. I believe the use of logistic regression for the post-diagnosis time window remains a methodological limitation, especially given the implied causal interpretation in certain parts of the manuscript. The authors have softened the language but have not changed the analytic approach.

In my reading, I believe this analytic approach could conflate directionality and does not account for variable follow-up time — which is crucial when asserting that downstream comorbidities might be outcomes of MI. A time-to-event model (e.g., Cox regression) or Poisson regression with offset for person-time would have been more appropriate.

Sensitivity analyses are helpful but do not adequately describe how they handle censoring or missing diagnoses outside the healthcare system

Additional minor comments:

Use of Vasectomy Patients as Controls - While well justified (fertile comparator), there remains potential bias due to different healthcare-seeking behaviors.

Multiple testing - The authors use Benjamini-Hochberg correction, but no power analysis or estimate of false positives from correlated phenotypes is provided.

Version 5:

Reviewer comments:

Reviewer #4

(Remarks to the Author)

The authors have addressed previous comments on methodology by adding analyses based on Cox regression. I think this is useful, but I agree with another reviewer that the design disregarding time should be mentioned as a limitation. Also a few more details on the Cox regression would be helpful, in particular how time was handled both with respect to the study design and the omission of the first 6 months.

Reviewer #5

(Remarks to the Author)

The authors have sufficiently addressed my prior comments but I think the Cox analysis should be featured in the main results rather than the supplement (due to the concerns with causality etc. for the logistic regression).

We would like to thank the reviewers for their constructive feedback. Please find the point by point responses below.

Referee expertise:

Referee #1: screening, biomedical informatics

Referee #2: male infertility

Reviewers' comments:

Reviewer #1 (Remarks to the Author):

Thank you for the opportunity to review the manuscript entitled “Leveraging Electronic Medical Records from Two Hospital Systems Identifies Male Infertility Associated Comorbidities Across Time”.

As context, I am experienced with EHR data and modeling thereof, but am neither an expert in infertility or in association studies using EHR data.

This manuscript describes a multi-site analysis of diagnoses identified following a male infertility diagnosis. It studies an important, understudied problem. The study is well-designed, the report is very clearly written, and the conclusions seem reasonable.

Thank you for recognizing the importance of studying male infertility.

However, in the context of the journal’s aim to publish findings that “represent significant advances in preventing, diagnosing, or treating human disease” and criteria for publication, it does not appear to me that this manuscript reports sufficient impact and/or innovation. The approach of using multiple sensitivity analyses and multiple systems’ data seems solid, but no claims are made regarding the novelty of the methods or approach. There are interesting associations identified that hold-up to sensitivity, multi-site, and clinical analyses; but, the proposed consequences are further study to better assess significance.

Thank you for recognizing the robustness of the methodology. We confirm that we do not claim novelty on the methods. This approach has been applied by both the Sirota and Aghaeepour labs in the context of AD and recurrent pregnancy loss.¹⁻³ Nonetheless, this approach has not been utilized for male infertility. While we identified

expected comorbidities such as abnormal spermatozoa, we also identified novel comorbidities associated with male infertility before the 6-month cutoff, such as hypothyroidism, prostate cancer, and other anemias. Our study design generates hypotheses that can be further explored via longitudinal analyses to test their clinical significance, which is currently out of the scope of this study. We have now clarified comorbidities of interest in the introduction section on page 3, lines 95-96: “these less-expected comorbidities include hypothyroidism, other anemias, and prostate cancer”. We have also now highlighted these more interesting and novel associations in the discussion on page 12, lines 498-516.

“However, we also found a number of less expected comorbidities, such as anemias, hypothyroidism, and prostate cancer. Prior studies have explored the relationship between male infertility with each of these less expected comorbidities to varying extents. The research on anemia generally is limited, although it is known that sickle cell disease, which can lead to primary and secondary testicular failure, can infer a higher risk for male infertility.²⁵⁻²⁷ Importantly, the current report did not identify sickle cell disease as the specific etiology.

For hypothyroidism, prior research has been primarily limited to rodent studies or human studies with a small number of participants.²⁸⁻³⁵ While thyroid dysfunction is a common cause of female infertility, the evaluation of the infertile male does not include such screening.³⁶ Finally, studies exploring the relationship between prostate cancer and male infertility often measure the risk of developing prostate cancer after receiving a male infertility diagnosis.³⁷⁻⁴¹ This is distinct from our finding, which suggests that patients with prostate cancer are receiving this diagnosis prior to or concomitant with their male infertility diagnosis, suggesting that it is possible that these patients are seeking treatment for their infertility, which could be the result of cancer treatment.⁴² The average age of prostate cancer diagnosis occurs in the seventh decade of life, an age beyond which reproduction is desired or common. Nevertheless, the age of paternity is increasing, and other groups have reported that reproduction should be increasingly considered among prostate cancer patients.^{43,44}”

Specific comments:

- How sensitive and specific is the MI diagnosis criteria applied?

Thank you for bringing up this important consideration. We cannot calculate sensitivity and specificity of the MI diagnosis criteria due to inherent lack of access to the ground

truth in these datasets. We have added the following in the discussion section to highlight this on page 14, lines 612-615:

“Moreover, the sensitivity and specificity of our criteria for male infertility and vasectomy are inherently incalculable in the UC and Stanford EHR due to lack of accessible ground truth. That is, we do not know how many patients truly have male infertility or have had a vasectomy in these databases.”

Generally, having broad inclusion criteria would maximize sensitivity while potentially compromising specificity. More stringent inclusion criteria generally will increase specificity while decreasing sensitivity. For example, a review in 2017 by Khandwala et al. demonstrated that of the patients who have undergone a semen analysis and have been diagnosed with at least one of five male infertility ICD-9 codes (606.0, 606.1, 606.8, 606.9, and V26.21), 92.4% were determined to have abnormal semen quality. If the patients had at least three distinct infertility codes (which could be construed as a “stricter” inclusion criteria), this rises to 99.8%.⁴ This is now included in the discussion on page 15, lines 615-621:

“In general, having broad inclusion criteria would maximize sensitivity while potentially compromising specificity, while more stringent inclusion criteria generally will increase specificity while decreasing sensitivity. For example, a review in 2017 by Khandwala et al. demonstrated that of the patients who have undergone a semen analysis and have been diagnosed with at least one of five male infertility ICD-9 codes (606.0, 606.1, 606.8, 606.9, and V26.21), 92.4% were determined to have abnormal semen quality. If the patients had at least three distinct infertility codes, this rises to 99.8%.”

- Who are the outliers in the UMAP in Figure 2? Cluster appears associated with EHR utilization? Expect these would affect the embedding space considerably. Why is further refinement of the data not called for?

Thank you for highlighting this. Indeed, the patients in the outlier cluster tend to have lower hospital utilization recorded in the EHR, particularly before the 6-month cutoff, as can be seen in Supplementary Figure 6. We accounted for this by performing the hospital utilization sensitivity analysis.

The patient outliers in the UMAP in Figure 2 also have fewer diagnoses before the 6-month cutoff date relative to non-outlier patients. We report this in new Supplementary Data 6. While we filtered for male infertility patients with at least one non male infertility diagnosis as well as for vasectomy patients with at least one diagnosis (see

Supplementary Figures 1 and 2) to account for patients with fewer records in the EHR, this did not remove all patients with few records.

- As discussed, even for findings that held-up in sensitivity analyses and across institutions, assessing clinical significance is challenging. The authors cite a need for longitudinal analyses. Are there further analyses that could be performed using available EHR data that could provide complementary support for these findings, such as time-to-event analyses, association using validated computable phenotypes for a comorbidity, or analyses indexed on the comorbidity.

Thank you for recognizing the importance of followup longitudinal analyses to assess clinical significance. The scope of our study was to identify male infertility associated comorbidities with a number of sensitivity analyses. A future direction of our work would indeed include longitudinal analyses that would help clarify the relationships between male infertility and associated comorbidities (causality, etc). For example, a retrospective cohort study can test whether patients in the EHR diagnosed with renal conditions are more likely to receive a MI diagnosis relative to patients who have not been diagnosed with renal conditions. We could also utilize, for instance, Cox proportional hazards models to assess whether patients with a MI diagnosis have a higher hazard rate for developing cardiovascular conditions relative to vasectomy patients. We now include this in the discussion section on page 13, lines 564-568:

“For example, a retrospective cohort study can test whether patients with renal conditions are more likely to develop MI, while Cox proportional hazards models can test, for instance, whether patients with MI are more likely to develop cardiovascular conditions relative to patients who have had a vasectomy.”

Reviewer #2 (Remarks to the Author):

The authors discuss the association between medical conditions and MI using 2 separate EMR databases to determine what conditions are associate w/ MI. Specifically looking at temporal relationships of diagnosis (within 6 months vs after). The outcomes are certainly meriting publication however the manuscript is very difficult to interpret in its current condition. There is a significant amount of data presented to the point it dilutes some of the important findings. For example, the discussing regarding the association of diabetes (which has been published on previously) overshadows the more novel findings of potentially missing different CV diagnoses. I would consider, condensing the results and discussion section to either focus on these findings OR focus on the amalgamation of diagnoses (expected vs unexpected in aggregate) rather than focus on the specific diagnosis.

Thank you for your feedback regarding how we can organize our results and discussion so that the main findings are clearer for the reader. We condensed our results through more concise reporting of UMAP findings and by focusing on the primary analysis findings, moving sensitivity analysis findings to supplementary figures and data. We also condensed the discussion regarding the associated comorbidities shared across all analyses and both hospital systems before the 6-month cutoff and focused our discussion on the less expected male infertility associated comorbidities (page 12, lines 496-524).

“However, we also found a number of less expected comorbidities, such as anemias, hypothyroidism, and prostate cancer. Prior studies have explored the relationship between male infertility with each of these less expected comorbidities to varying extents. The research on anemia generally is limited, although it is known that sickle cell disease, which can lead to primary and secondary testicular failure, can infer a higher risk for male infertility.²⁵⁻²⁷ Importantly, the current report did not identify sickle cell disease as the specific etiology.

For hypothyroidism, prior research has been primarily limited to rodent studies or human studies with a small number of participants.²⁸⁻³⁵ While thyroid dysfunction is a common cause of female infertility, the evaluation of the infertile male does not include such screening.³⁶ Finally, studies exploring the relationship between prostate cancer and male infertility often measure the risk of developing prostate cancer after receiving a male infertility diagnosis.³⁷⁻⁴¹ This is distinct from our finding, which suggests that patients with prostate cancer are receiving this diagnosis prior to or concomitant with their male infertility diagnosis, suggesting that it is possible that these patients are seeking treatment for their infertility, which could be the result of cancer treatment.⁴² The average age of prostate cancer diagnosis occurs in the seventh decade of life, an age beyond which reproduction is desired or common. Nevertheless, the age of paternity is increasing, and other groups have reported that reproduction should be increasingly considered among prostate cancer patients.^{43,44}

We also found several circulatory system diagnoses that were positively associated with MI before the 6-month cutoff in the UC analysis, including circulatory disease, congestive heart failure, and hypertensive complications, with one study suggesting an association between hypertension and impaired semen quality.⁴⁵ Our results suggest that it may be beneficial to perform larger, longitudinal studies to clarify the relationship between MI and these

comorbidities, which are all treatable conditions. Treatment for hypothyroidism and anemia in particular may be promising for restoring male fertility.”

Abstract:

The wording is awkward in lines 21 and 28-32, consider revising for clarity.

Line 21 - Consider stating MI “solely” can be attributed to 30% of the cases of infertility or discuss that MI may be implicated in up to 50% (in tandem with female factors) à as discussed in the introduction.

Thank you for your feedback on wording in the abstract.

We have revised line 21 in the abstract. The revised line is on page 1, lines 23-24:

“15% of those attempting to conceive experience infertility. Male infertility (MI) is the sole cause of 20-30% of infertility cases, and it is a contributing factor for an additional 15-20% of cases.”

We have also revised lines 28-32 for clarity. The revised lines are on page 1-2, lines 41-50:

“Results

Here, we identify 15 diagnoses that are positively associated with MI before the 6-month cutoff across both hospital systems and all analyses, including less expected comorbidities such as hypothyroidism and other anemias. We also find that abnormal spermatozoa and testicular hypofunction are positively associated with MI after the 6-month cutoff across both hospital systems and all analyses.

Conclusions

Our findings set the groundwork for future studies to clarify the relationship between less expected comorbidities and MI.”

Discussion:

Lines 363-379 state that the relationship between DM and MI has “not been well-established” but cite 10 papers discussing the relationship, would consider removing this statement.

Thank you for the feedback. We have removed the statement that the relationship between DM and MI has “not been well-established.”

Results:

I appreciate the transparency of this section. I would strongly consider moving a portion of the results to the methods and/or even a supplemental methods section to improve the readability. For example, lines 87-94 describe the methodology and then are followed by the results.

Likewise, lines 155-158 summarize the findings from the aforementioned several paragraphs and would be more concise for readers.

Thank you for suggesting that we consider condensing the results and moving some portion to the methods.

We have moved “Low-dimensional embedding takes as input an $m \times n$ matrix, where each row (m_i) represents a patient, and each column (n_j) represents a particular diagnosis. The column values indicate whether the patient has or does not have the particular diagnosis using the binary values 1 or 0, respectively. UMAP outputs an $m \times 2$ matrix from the $m \times n$ matrix, which allows us to visualize patients’ diagnosis profiles in 2 dimensions.” to the methods section, which can be found on page 4, lines 146-151.

We also condensed the results of the UMAP section that lines 155-158 summarizes; this can be found on page 7, lines 291-306.

Methods:

There needs to be a discussion of what conditions are expected and what conditions are not expected based on the literature. For example DM is considered “not expected” though the relationship has been published on significantly i.e. PMIDs 25814158, 29887834, 26674559.

Thank you for suggesting that we clarify what conditions are not expected based on the literature. For diagnoses significantly associated before the 6-month cutoff across all analyses and both hospital systems, we focused on anemia, hypothyroidism, and prostate cancer as unexpected comorbidities. While it may seem like prostate cancer, and particularly its treatment, would clearly be associated with male infertility, we thought that this finding was interesting due to this diagnosis arising prior to or concomitant with male infertility. Discussion of these findings can be found on page 12, lines 498-516.

“However, we also found a number of less expected comorbidities, such as anemias, hypothyroidism, and prostate cancer. Prior studies have explored the relationship between male infertility with each of these less expected

comorbidities to varying extents. The research on anemia generally is limited, although it is known that sickle cell disease, which can lead to primary and secondary testicular failure, can infer a higher risk for male infertility.²⁵⁻²⁷ Importantly, the current report did not identify sickle cell disease as the specific etiology.

For hypothyroidism, prior research has been primarily limited to rodent studies or human studies with a small number of participants.²⁸⁻³⁵ While thyroid dysfunction is a common cause of female infertility, the evaluation of the infertile male does not include such screening.³⁶ Finally, studies exploring the relationship between prostate cancer and male infertility often measure the risk of developing prostate cancer after receiving a male infertility diagnosis.³⁷⁻⁴¹ This is distinct from our finding, which suggests that patients with prostate cancer are receiving this diagnosis prior to or concomitant with their male infertility diagnosis, suggesting that it is possible that these patients are seeking treatment for their infertility, which could be the result of cancer treatment.⁴² The average age of prostate cancer diagnosis occurs in the seventh decade of life, an age beyond which reproduction is desired or common. Nevertheless, the age of paternity is increasing, and other groups have reported that reproduction should be increasingly considered among prostate cancer patients.^{43,44}

References:

1. Tang, A. S. *et al.* Deep phenotyping of Alzheimer's disease leveraging electronic medical records identifies sex-specific clinical associations. *Nat. Commun.* **13**, 675 (2022).
2. Woldemariam, S. R., Tang, A. S., Oskotsky, T. T., Yaffe, K. & Sirota, M. Similarities and differences in Alzheimer's dementia comorbidities in racialized populations identified from electronic medical records. *Commun. Med.* **3**, 1–14 (2023).
3. Roger, J. *et al.* Leveraging electronic health records to identify risk factors for recurrent pregnancy loss across two medical centers: a case-control study. Preprint at <https://doi.org/10.21203/rs.3.rs-2631220/v1> (2023).
4. Khandwala, Y. S., Zhang, C. A., Li, S., Cullen, M. R. & Eisenberg, M. L. Validity of Claims Data for the Identification of Male Infertility. *Curr. Urol. Rep.* **18**, 68 (2017).

Reviewer #1 (Remarks to the Author):

Thank you for the opportunity to review the resubmission of the manuscript entitled “Leveraging Electronic Medical Records from Two Hospital Systems Identifies Male Infertility Associated Comorbidities Across Time”. The revisions have improved readability and interpretability considerably. I expect the approach and findings will be interesting to the readership. I provide the below suggestions to further highlight findings most useful for readership.

We would like to thank the reviewer for taking the time to re-review the manuscript and are glad that the revisions have improved the paper. Please see point by point responses below.

Comments:

- All the different analyses provide insight into the quality and interpretability of the study. However, the presentation is very complicated for assessing any individual clinical question. If the goal is to ‘set the groundwork for future studies to clarify the relationship between less expected comorbidities and MI’, consider presenting a view of the data relative to specific clinical hypotheses rather than across all clinical hypotheses. Consider highlighting some of the less expected comorbidities by pulling together a set of the analyses (e.g. ORs and p-values for several sub-analyses) together into a single view/table. This would help in assessing the significance of diagnoses/diagnosis groups highlighted in the discussion. If this frame is buried in a supplemental table, then I have missed it.

We would like to thank the reviewer for the suggestion. The goal is indeed to set the groundwork for future studies to clarify the relationship between less expected comorbidities and MI. To this end, we made a new Table 2 highlighting the 15 diagnoses significantly associated with male infertility before the 6 month cutoff across all analyses and both institutions:

phenotype	UC odds ratio	UC p-value	Stanford odds ratio	Stanford p-value
Abnormal spermatozoa	23.61	6.52E-105	40.44	1.42E-08
Anterior pituitary disorders	7.92	1.62E-05	10.20	2.08E-02
Cancer of prostate	3.70	4.54E-05	5.48	1.52E-02
Decreased libido	2.13	4.46E-09	2.00	2.34E-02
Elevated prostate specific antigen [PSA]	1.97	5.15E-03	3.43	6.49E-04

Erectile dysfunction [ED]	2.28	1.30E-37	1.54	3.22E-04
Hyperplasia of prostate	2.08	5.48E-08	1.76	1.38E-02
Hypothyroidism NOS	1.97	1.15E-06	1.65	1.58E-02
Infertility, female	48.12	2.20E-06	26.67	3.31E-22
Other anemias	1.83	3.34E-07	2.45	8.43E-07
Other disorders of testis	9.22	1.83E-11	9.47	4.85E-10
Pituitary hypofunction	5.90	1.99E-06	25.80	1.98E-02
Testicular hypofunction	10.99	3.23E-229	13.84	1.14E+00
Type 2 diabetes	2.31	9.71E-13	1.76	4.69E-03
Varicose veins	5.20	1.15E-113	5.93	4.73E-20

Table 2: Diagnoses positively associated with male infertility prior to six months after receiving a male infertility diagnosis across all analyses and both hospital systems.

Phenotype corresponds to diagnosis. Diagnoses positively associated with male infertility before the six-month cutoff across the primary analysis and both sensitivity analyses at UC and Stanford. Shown are the odds ratios and p-values from the primary analyses. UC = University of California.

- Please include a limitation referring to the selection of control population. The motivation for selecting this control population is well described. But, there could be other differences between cases and controls that are causally associated with the observed outcomes but are not due to MI. Table 1 shows many differences, notably include race identified as Asian and utilization. Sensitivity analyses appear reasonably designed to try to account for this, but do not account for all potential bias.

We would like to thank the reviewer for this suggestion. We have added several sentences to the Discussion section on page 15, lines 634-636 on the limitation of selecting the appropriate control population:

“there could be differences between MI patients and vasectomy patients that are causally associated with the observed outcomes but are not due to MI. While our sensitivity analyses try to account for this, they likely do not account for all potential bias.”

- Within 6 months, why does it make sense to observe just as many significant negative associations as significant positive associations given the design and the observation in the post-6 month data? Does this not suggest that many of the positive associations are likely false-positives?

Thank you for pointing out the potential for false positives. We attempted to address this concern by running two sensitivity analyses - the social determinants of health sensitivity analysis and the hospital utilization sensitivity analysis - to account for potential confounding factors that could affect the significance of associations between diagnoses and male infertility. We also ran a validation study using Stanford's EHR to assess which findings were consistent across institutions. We think running these sensitivity analyses and validating associations across two different hospital systems help mitigate the possibility that findings consistent between all of these analyses are false positives.

We addressed the possibility of false positives and attempts to mitigate them by adding the following to the Discussion section on page 15, lines 644-647:

"It is also possible that some of our findings are false positives, which we attempted to mitigate by identifying significant male infertility associated diagnoses not only in the primary analysis, but also in the social determinants of health and hospital utilization sensitivity analyses across both UC and Stanford."

- Improve resolution of supplemental figures.

Thank you, we have improved the resolution of the supplementary figures.

- Code repos is not accessible

The code is now accessible using the following url:

https://github.com/SarahRoWo/Logistic_Regression_Python_MI_Paper

- Please explain why it is not preferable to use apply a common UMAP transformation across analyses to facilitate comparisons, such as between UC and Stanford.

Thank you for asking this question. We applied UMAP visualizations for each cohort independently. While we were able to apply visualizations across UC institutions, we were not able to apply a common UMAP transformation across analyses at UC and Stanford. This is due to data sharing restrictions; specifically, we cannot share patient-level data between UC and Stanford, including UMAP coordinates.

Reviewer #2 (Remarks to the Author):

The updated manuscript is much more readable in its current state and emphasizes potentially novel findings surrounding anemia, thyroid dysfunction, and male infertility.

We would like to thank the reviewer for recognizing the improvements in our work.

I challenge the association between prostate cancer and the development of male factor infertility, being significant only due to iatrogenic factors. Is there a way to control for surgery, radiation, and androgen deprivation in this cohort? If not, I would either remove the association given these significant limitations or state this limitation explicitly.

We would like to thank the reviewer for bringing up this point. We certainly agree that prostate cancer (as with many other types of) treatment can lead to male infertility. Unfortunately, the current data does not allow us to control for surgery, radiation and androgen deprivation treatment. However, we added this important point and note that future studies are required (page 13, lines 556-559).

“However, to assess whether the association between prostate cancer and the development of MI could indeed be due to cancer treatment, confounders such as surgery, radiation, and androgen deprivation would need to be controlled for and hopefully will be explored in future studies.”

Reviewer #3 (Remarks to the Author):

The authors present a data drive approach to identify whether Male Infertility (MI) compared to vasectomy (health controls) was associated with non-MI disease diagnoses. They specifically focus on non-MI diagnoses between 0 and 6 months after an MI/VAS diagnosis and, non-MI diagnoses after the 6 months period.

We would like to thank the reviewer for summarizing our work.

My comments are as follows.

1. The logistic regression analyses use MI/VAS as the outcome variable, this should be the exposure/independent variable. Using MI/VAS as the exposure variable ensure the odds ratios can be readily interpreted which provide information on the magnitude of association.

Thank you for the feedback. We chose MI/VAS as the outcome variable for our logistic regression models, as our objective was to identify diagnoses related to MI (or VAS), not the predictive value of MI/VAS on each particular diagnosis. Moreover, using logistic regression to assess the predictive value of MI/VAS on each particular diagnosis in our study has significant limitations. In particular, since many diagnoses explored are relatively rare, maximum likelihood estimation convergence failures and singular matrices can occur for logistic regression models that use MI/VAS as an exposure variable.

2. In line with the previous comment please report the odds ratio in a supplementary table (for the primary and sensitivity analyses). Why do the figure plot the $\log_{10}(\text{odds ratio})$, the natural logarithm of the odds ratio (the regression coefficients of a logistic regression model) is more sensible.

Thank you for highlighting the importance of reporting the odd ratios for each logistic regression analysis. These are reported in supplementary data files 8 (results for primary and sensitivity analyses before the 6 month cutoff for UC), 9 (results for primary and sensitivity analyses before the 6 month cutoff for Stanford), 14 (results for primary and sensitivity analyses after the 6-month cutoff for UC), and 15 (results for primary and sensitivity analyses after the 6-month cutoff for Stanford) under the column `odds_ratio_has_phenotype`.

Additionally, thank you for suggesting that we use the natural log of the odds ratio for our plots. We chose $\log_{10}(\text{odds ratio})$ for ease of interpretability independent of the regression coefficients of our analyses. However, we acknowledge that using the natural log of the odds ratio instead would allow readers to ascertain the regression coefficients associated with each diagnosis. Therefore, we changed the axes from $\log_{10}(\text{odds ratio})$ to $\ln(\text{odds ratio})$.

3. While I understand the desire to look at short/medium and long-term associations, the nature of the EHR data make it highly likely that diagnoses within the 0 to 6 months period were already present before the MI diagnosis. This of course very much depends on the health seeking behaviour of a participant, which clearly will be distinct for MI patients (these will be more thoroughly check than VAS). As such differences in non-MI disease diagnoses are to be expected, especially within this short/medium term follow-up period. Taking a more biological approach how would the authors expect MI would lead to de novo disease development such as the onset of heart failure (figure 3.a)? This is not to say that the analysis is wrong, it should just be extremely clear that this analysis is likely picking up pre-existing conditions.

We would like to thank the reviewer for bringing up these important points.

We agree that diagnoses first obtained within 6 months of a MI diagnosis may have been present before the MI diagnosis. We did not mean to imply that MI would cause de novo disease but rather be a marker for it, perhaps due to a shared genetic or exposure etiology. Our logistic regression models explored diagnoses before the 6-month cutoff, not within the 6-month cutoff. This was a descriptive error on our part, and we apologize for the confusion. We replaced the phrase “within 6 months” with “prior to 6 months” after receiving a MI diagnosis throughout the manuscript to reflect this. In this context, the earliest diagnoses would have been first observed in the UC EHR in 2012. We also clarified the purpose of the 6-month cutoff in the Introduction section on page 3, lines 94-100:

“Here, we performed a temporally-stratified case-control study across two independent hospital systems to identify MI-associated comorbidities. We explored conditions first diagnosed before 6 months after a MI diagnosis or vasectomy-related record to identify comorbidities that may have been present prior to a MI diagnosis and thus could be risk factors for MI. We also explored conditions first diagnosed at least 6 months after a MI diagnosis or vasectomy-related record to identify comorbidities that may have manifested after a MI diagnosis and thus could be adverse outcomes of MI.”

4. The analysis looking at outcomes after 6 months is difficult to interpret due to the open ended follow-up period. Did the outcome occur 7 months after MI diagnosis or 7 years? Possibly the analysis can be enriched by providing the median and 1st and 3rd quartiles of the diagnosis time relative to the VAS/MI diagnosis?

We would like to thank the reviewer for bringing up this important point. We now include the median, 1st and 3rd quartiles of the diagnosis time relative to the first male infertility diagnosis or vasectomy record for each association in Supplementary Data 7.

We also added the following to the Results section on page 9, lines 355-356:

“The distribution of timing between each diagnosis and first MI diagnosis or vasectomy-related record can be found in Supplementary Data 7.”

5. In fact a logistic regression model (which uses a Binomial distribution) is invalid when the follow-up time is not fixed. So please fix the follow-up time to something sensible (say 5/10 years).

Thank you for the comment. We agree that fixed follow-up times are important when testing whether male infertility is predictive of a particular diagnosis. However, we think that how we structured our logistic regression models is appropriate for our study, which aimed to identify diagnoses associated with male infertility. Such association analyses do not require survival analysis methods.

That said, we ran fixed follow-up time analyses for patients with at least 12, 24, 36, 48, or 60 months of follow-up time and explored diagnoses obtained within these times. Supplementary Data 19 contains results for UC, while Supplementary Data 20 contains results for Stanford.

Generally, from these analyses, we found that the diagnoses enriched for MI patients are concentrated in three disease categories at UC: genitourinary, endocrine/metabolic, and circulatory system. This is most apparent when 24 or 36 months are used for fixed follow-up time and generally consistent with findings for the after 6-month cutoff analyses that were performed without a fixed follow-up time. By contrast, at Stanford, only Testicular hypofunction was significantly associated with MI for the fixed follow-up time analyses, with the exception of the 60-month fixed follow-up time analyses, in which there were no diagnoses significantly associated with MI.

We also added the following to the manuscript:

Methods, page 5, lines 196-198: “We ran the above three analyses for patients with no fixed follow-up time, as well as for patients with at least 12, 24, 36, 48, or 60 months of follow-up time. For fixed follow-up times, we explored diagnoses obtained within those timeframes.”

Methods, page 6, lines 242-244: “As with UC, we ran the above three analyses for patients with no fixed follow-up time, as well as for patients with at least 12, 24, 36, 48, or 60 months of follow-up time. For fixed follow-up times, we explored diagnoses obtained within those timeframes.”

Results, page 11, lines 438-447: “We also assessed patients with at least 12, 24, 36, 48, or 60 months of follow-up time beyond 6 months of first being diagnosed with MI or receiving a vasectomy-related record, and we tested for MI-associated comorbidities diagnosed within those timeframes. The number of patients lost for each follow-up cutoff

time can be found in Supplementary Data 18 for UC and Stanford. At UC, genitourinary, endocrine/metabolic, and circulatory system diagnoses were enriched for MI patients in the primary analyses. This is most apparent for patients with at least 24 or 36 months of follow-up (Supplementary Data 19). For Stanford, only testicular hypofunction was significantly associated with MI for all fixed follow-up time analyses, with the exception of the 60-month fixed follow-up analyses, in which no significantly associated diagnoses were found (Supplementary Data 20).”

6. In line with this follow-up time comment, after settling on a fixed time period, please indicate how many participants were lost to follow-up?

- a. Below is a table indicating the number of patients lost for a number of fixed time periods at UC. Note that without fixed follow-up time, the number of male infertility patients = 6,531, while the number of vasectomy patients = 8,353.

follow-up cutoff time (months)	male infertility patients lost to follow-up (n)	vasectomy patients lost to follow-up (n)
12	3,554	4,609
24	4,282	5,522
36	4,790	6,301
48	5,219	6,877
60	5,578	7,363

- b. Below is a table indicating the number of patients lost for a number of fixed time periods at Stanford. Note that without fixed follow-up time, the number of male infertility patients = 5,551, while the number of vasectomy patients = 2,464.

follow-up cutoff time (months)	male infertility patients lost to follow-up (n)	vasectomy patients lost to follow-up (n)
12	1,920	1,139
24	2,448	1,484
36	2,881	1,734
48	3,292	1,916
60	3,605	2,054

As noted in response to question 5, we added the following to the manuscript in the Results section on page 11, lines 440-441: “The number of patients lost for each follow-up cutoff time can be found in Supplementary Data 18 for UC and Stanford.”

7. Depending on the amount of censoring more formal survival methods such as Cox's regression model might be relevant. Such a model would be able to natively account for censoring.

Thank you for the comment. We agree that performing Cox regression would be highly relevant for our work. Unfortunately, it is out of the scope of the current study, as our main objective was to identify diagnoses associated with male infertility, not to test whether a male infertility diagnosis is predictive of developing a particular diagnosis. However, we hope that we can perform Cox regression in a future study to clarify relationships between male infertility and the associated diagnoses identified in this study.

8. Please clarify what would have happened if a participant got a diagnosis in a different hospital. Would this diagnosis be registered in the currently used data? If not, how can the researcher be sure about the exposure and outcome status?

We would like to thank the reviewer for bringing up this question. While health information can be shared through a Health Information Exchange (e.g., "Care Everywhere"), under federal privacy law (Health Insurance Portability and Accountability Act, "HIPAA"), it is not legal to use data from another institution for research outside of patient treatment, and UCSF IRB protocol approvals do not cover this outside data. While prior diagnoses that patients share with our clinicians may be recorded in our EHR, ones that are not divulged would be missed. Moreover, the EHR does not necessarily note which diagnoses were obtained outside of the corresponding institution. This missingness is an inherent limitation of studies utilizing EHR data, and thus, we cannot be absolutely sure about the exposure and outcome status of diagnoses.

We added the following to highlight this specific limitation in the Discussion section on page 15, lines 639-642:

"We are also unable to tell whether they were first diagnosed outside of UC or Stanford due to the legal limitations placed on systematically sharing personal health information across institutions; this potential missingness is an inherent limitation of studies utilizing EHR data."

9. While I appreciate the UMAP analysis, is it really that surprising that people with MI have a different diagnosis profile when compared to people with VAS? MI people get screened while VAS do not get screened – this alone would ensure there is a difference in diagnosis.

We would like to thank the reviewer for bringing up this point. We use UMAP as a visualization tool to demonstrate what we expect to see - a separation of cases and controls. Moreover, while directed questions may be slightly different, both MI and VAS evaluations mandate a thorough history and physical examination to evaluate for comorbidities. The guideline-based evaluation for men presenting with MI or VAS mandate a thorough history to help understand the etiology of the condition (MI) or suitability for a procedure (VAS). While some conditions may be related to a MI evaluation (e.g. varicocele, endocrine disorder), many others (e.g. heart disease) would be unlikely to be diagnosed during an MI encounter. Nevertheless, the point is well taken and we acknowledge differences in clinical pathways for each and have added such language to the Discussion section on page 15, lines 627-634:

“First, some of the differences observed in the UMAP of diagnosis profiles between MI patients (cases) and vasectomy patients (controls) might be due to different screening practices for MI diagnoses and vasectomy procedures. Evaluations for both MI and vasectomy, however, mandate a thorough acquisition of medical history as well as a physical examination to evaluate for comorbidities. While some conditions may be related to a specific screening practice, such as identifying varicoceles during a MI evaluation, many others, such as hypothyroidism, would be unlikely to be diagnosed during a MI evaluation.”

Overall the figure are nicely formatted.

Thank you.

Figure 3.c uses a piece wise y-axis, this may well be erroneously interpreted. Please use a normal continuous y-axis, if needed one could consider truncating the y values above a certain threshold. This truncation should of course be explained and the un-truncated values presented someplace (e.g. in the figure note if there are not too many)

Thank you, we have truncated Figure 3c.

Figure 3.d, Figure 4 c-d need y-axis labels.

Thank you, this was corrected.

Figure 7, the resolution is too low.

Thank you, we improved the resolution.

Please find the point by point responses below:

Reviewer 1:

Thank you for your positive feedback.

Reviewer 3:

I am grateful to the authors for their detailed response. The authors now indicate that any diagnoses made in a different hospital were not recorded, as such the exposure and outcome variables are incorrect, invalidating their analysis and the present manuscript. The authors remark that EHR data always suffers from this type of incomplete record keeping is of course incorrect, countries like Estonia or the UK have national coverage severely limiting this issue. The current problem is so fundamental that even if the comments below could be addressed the study is still faulty beyond repair.

Thank you for the feedback. We respectfully but fundamentally disagree with the harsh and unjustified statement that the study is “faulty beyond repair”. We want to be clear: we did not say that any diagnoses made in a different hospital are not recorded. To quote our response from the most recent round of responses to reviewer comments, “While prior diagnoses that patients share with our clinicians may be recorded in our EHR, ones that are not divulged would be missed.” While we acknowledge that patients who obtain care at multiple facilities may not have a complete record at any particular one, we would like to clarify that such medical promiscuity is relatively uncommon. Importantly, most significant diagnoses (e.g. diabetes, coronary artery disease, etc.) do follow patients between clinical environments. Furthermore, if there are missing diagnoses, they are likely equally distributed among patients with and without male infertility. Such nondifferential misclassification would be expected to bias results to the null. Thus, any significant results may even be larger than calculated.

We would also like to take the opportunity to clarify in the manuscript that EHR data in the United States often have incomplete record keeping, and we acknowledge that the national coverage that exists in other countries mitigates this limitation. That said, numerous prior studies have utilized data-driven approaches on EHR data derived from US hospitals; this includes our prior published work, which utilize University of California health centers (see references 14: Tang et al., published in Nat Commun in 2022; <https://doi.org/10.1038/s41467-022-28273-0> as well as 16: Woldemariam et al., which was published here in Commun Med in 2023; <https://doi.org/10.1038/s43856-023-00280-2>). Of course, complete record keeping similar to several European countries would be ideal, but it is just not possible in the US.

We also observe that many of our results are supported by prior studies, and these have been highlighted extensively in our paper (see references 10-13, 23, 28-41, and 45-55 in the manuscript). This suggests that our approach can identify meaningful findings that are consistent with studies that utilize different approaches that do not have the same limitations.

Furthermore, while the authors employ a binomial regression model which fundamentally assumes equal follow-up time, with anybody with a shorter follow-up time (and without an event) should be treated as having missing data, the authors choose to ignore this and simply assign these people to being free of disease. This is completely incorrect.

Respectfully, we did not ignore the concern about having equal follow-up times, and we extensively addressed this in the most recent round of reviews. We performed logistic regression on patients with fixed follow-up times of 1, 2, 3, 4, or 5 years. We discussed the results from these analyses in the Results section on page 11, lines 438-447, and we shared the complete results in Supplementary Data files 18-20. The logistic regression model assumes the data are missing at random and we investigated this assumption in our fixed follow-up time analyses (Results section on page 11).

If the authors want to use the full follow-up either using imputation to deal with people that have shorter follow-up and apply a binomial model, or use a time-to-event approach that allows for censoring.

We would like to thank the reviewer for the suggestion. We have previously addressed why we did not use a time-to-event approach in the previous round of reviews. While we agree that a time-to-event approach that allows for censoring would be critical if we were asking whether there is predictive value of a male infertility diagnosis on subsequently receiving a particular diagnosis, we are instead asking about associations between male infertility and diagnoses. Logistic regression is thus appropriate to use for such an association study. Using a survival analysis, which would be a Cox proportional hazards model here, makes stricter assumptions about the data and about a treatment effect in predicting an outcome. Those assumptions are not appropriate to the hypothesis of associations within the context of EHR data analysis.

We address the possibility that differences in follow-up may bias the results by performing a hospital utilization sensitivity analysis, which captures both months in the EHR as well as number of visits both before and after the 6-month cutoff used in our analyses.

In order to further ensure the robustness of our statistical approaches, we have invited a biostatistician, Dr. Isabel Elaine Allen (cc'ed), to review our approach in the second round of reviews. She was supportive of our work, worked with us on sensitivity analyses with respect to fixed follow-up times to strengthen our conclusions, and joined the manuscript as a co-author.

In part due to the above issue, the authors use the outcome as the dependent variable and the exposure as an independent variable. As indicated before this is incorrect, and prohibits straightforward interpretation of the results.

We respectfully disagree with the above comment by the reviewer. In our association study, the outcome variable is the dependent variable. We have also extensively addressed this in the previous round of reviews, which might have gotten missed by the reviewer. We chose male infertility (MI) / vasectomy (VAS) as the outcome variable for our logistic regression models, as our objective was to identify diagnoses related to MI (or VAS), not the predictive value of MI/VAS on each particular diagnosis. Moreover, using logistic regression to assess the predictive value of MI/VAS on each particular diagnosis in our study has significant limitations. In particular, since many diagnoses explored are relatively rare, maximum likelihood estimation convergence failures and singular matrices can occur for logistic regression models that use MI/VAS as an exposure variable. We have also consulted with Dr. Isabel Elaine Allen regarding the appropriateness of using the outcome variable as the dependent variable, given the hypotheses and objectives of our study, and she agrees that our reasoning is valid.

The authors reply stating that the diagnostic work-up for people with MI and male infertility is equal is clearly incorrect. It seems highly unlikely that people with VAS for example routinely receive an ECG or even the biochemistry would be distinct. As such even when there would be complete data, including information from other hospitals or GPs, such a comparison is biased due to differences in diagnostic procedure.

We respectfully disagree with the reviewer on this comment. If the critique concerned the comparison of the diagnostic work-up for people with male infertility (MI) vs VAS specifically, then we'd like to be clear: we did not say that the diagnostic work-up for people with MI vs VAS is equal. Here is our response from the previous round of reviews, which includes a quote from the Discussion:

Moreover, while directed questions may be slightly different, both MI and VAS evaluations mandate a thorough history and physical examination to evaluate for comorbidities. The guideline-based evaluation for men presenting with MI or VAS mandate a thorough history to help understand the etiology of the condition (MI) or suitability for a procedure (VAS). While some conditions may be related to a MI evaluation (e.g. varicocele, endocrine disorder), many others (e.g. heart disease) would be unlikely to be diagnosed during an MI encounter. Nevertheless, the point is well taken and we acknowledge differences in clinical pathways for each and have added such language to the Discussion section on page 15, lines 627-634:

“First, some of the differences observed in the UMAP of diagnosis profiles between MI patients (cases) and vasectomy patients (controls) might be due to different screening practices for MI diagnoses and vasectomy procedures. Evaluations for both MI and vasectomy, however, mandate a thorough acquisition of medical history as well as a physical examination to evaluate for comorbidities. While some conditions may be related to a specific screening practice, such as identifying varicoceles during a MI evaluation, many others, such as hypothyroidism, would be unlikely to be diagnosed during a MI evaluation.”

Our response is informed by the clinical expertise of one of our senior authors, Michael Eisenberg, MD, who is a Professor of Urology at Stanford University, Director of the Male Reproductive Medicine and Surgery program at Stanford (cc'ed), and male infertility specialist. He regularly sees patients who may have male infertility as well as patients who are considering a vasectomy. Thus, he has first-hand knowledge of how the diagnostic work-up of these patient groups compare. Indeed, the evaluation of both is thorough to assess the etiology of male infertility but also for the safety of a surgical procedure. Furthermore, we highlight a prior study that compared male infertility and vasectomy patients due to similarities in patient characteristics, so we think that vasectomy patients are a reasonable control group (see reference 12 in the manuscript: Brubaker et al.; [10.1111/andr.12436](https://doi.org/10.1111/andr.12436)).

Generally, observational case-control studies that utilize EHR data cannot be assumed to have equal diagnostic work-up between cases and controls. Nonetheless, numerous case-control studies have been accepted or published that utilize EHR data (see previously mentioned references 14 and 16 in the manuscript, as well as Ramey et al. (<https://doi.org/10.1101/2024.05.02.24306649>), which has recently been accepted for publication).

We also want to note that we did not assert that the diagnostic work-up for people with MI (which we assume the reviewer means myocardial infarction given the mention of ECG in the subsequent sentence) and male infertility is equal, as myocardial infarction was not mentioned in the manuscript. This could be a potential error or typo by the reviewer.

Reviewer #3 (Remarks to the Author):

I am grateful to the authors for raising this appeal, clearly I was mistaken in the comparison (fully my fault for paying insufficient time on this). Their letter has helped with this.

Thank you, we are happy to hear that the letter provided additional clarity.

Two things I would like to raise, one suggestion and one remaining confusion.

The suggestion: it seems the authors design might best be described as a nested case-control study, nested within a cohort of two hospital and where exposure variables are collected through a mixture of hospital records and self-reported data. It would help a lot if the authors could describe the study along these lines and explain where and when the study deviated from this design.

Thank you for the suggestion. We agree with this description and added the following to the Introduction on page 3, lines 97-100: "Here, we performed an observational case-control study utilizing structured EHR to identify MI-associated comorbidities retrospectively and prospectively. The cases and controls in our study were nested and compared within and between two independent hospital systems to identify these associations."

The part I am still confused about. The paper conducts two analysis, a retrospective analysis and a prospective analysis. In the retrospective analysis the authors indeed identify risk factors for male infertility. In the prospective analysis the authors say they are trying to "identify comorbidities that may have manifested after a MI diagnosis and thus could be adverse outcomes of MI". This implies that MI is an exposure and the diagnosis after MI are considered outcomes – which are subject to missing data due to shorter follow-up or because some diagnoses occurred in a different healthcare settings. Why do the authors still use MI as the outcome variable in their logistic regression analysis? A time to event analysis would at least deal with people having less follow-up (i.e., people with 2 and 4 year follow-up could be combined, rather than setting the person with 2 years follow-up at missing)? Furthermore for the prospective analysis there seems to be a much larger problem with outcomes not being registered in the two hospital. I do agree that for the retrospective analysis this is much less of a concern and the employed analysis makes sense.

Thank you for your question, and we apologize for the potential confusion. We employed an observational case-control study, where we explored patient records retrospectively and prospectively to identify male infertility associated comorbidities. We

did not design formal retrospective and prospective analyses, which would indeed identify risk factors and adverse outcomes of male infertility. Rather, we are *generating hypotheses* for what these risk factors and adverse outcomes of male infertility *could be* through these association analyses. In this context, for the analysis exploring patient records prospectively, we are asking whether patients with a given diagnosis have a higher probability of male infertility relative to those who do not. If they do, we take this to mean that the diagnosis and male infertility are associated, but *we do not prescribe any direction to the association*.

To clarify that we are not identifying risk factors or adverse outcomes but rather *potential* risk factors and adverse outcomes, we modified the following portions of text that discusses the identification of *potential* risk factors or adverse outcomes:

1. Plain language summary, page 2, lines 62-64: “Here, we applied a data-driven approach utilizing electronic health records from UC and Stanford to identify associated conditions that may be risk factors for or adverse outcomes of MI.”
2. Introduction, page 3, lines 100-105: “We explored conditions first diagnosed before 6 months after a MI diagnosis or vasectomy-related record to identify comorbidities that may have been present prior to a MI diagnosis; thus, these associated comorbidities could be risk factors for MI. We also explored conditions first diagnosed at least 6 months after a MI diagnosis or vasectomy-related record to identify comorbidities that may have manifested after a MI diagnosis; thus, these associated comorbidities could be adverse outcomes of MI.”
3. Discussion, page 17, lines 712-713: “Our temporally-stratified association analyses point to conditions that we hypothesize may be risk factors for or adverse outcomes of MI.”

Reviewer #4 (Remarks to the Author):

As raised by another reviewer, it is not entirely clear how time is taken into account in the design and analysis. Ideally it would seem that a nested case-control study taking time into account could have been designed, which would be followed by a time-to-event analysis. It seems that the authors have considered a nested case-control-type design which disregards time, and then (appropriately) used logistic regression for the analysis. It may be though that because of the structure of the underlying data patients with and without male infertility contribute different amounts of time. Has this been taken into account?

Thank you for the question. Indeed, we utilized a nested case-control study design that does not take time into account and thus used logistic regression for analysis. And we agree that, due to the structure of the underlying data, patients with and without male infertility may contribute different amounts of time. Therefore, to take this into account, ran Cox proportional hazards models on the 13 diagnoses found to be significant after the 6-month cutoff at the University of California for the primary, social determinants of health, and hospital utilization logistic regression analyses. Similar to the logistic regression analyses, we ran 3 Cox regression analyses for each diagnosis by adjusting for the following covariates:

1. primary analysis: male infertility status + age when first received male infertility diagnosis or vasectomy-related record + UC location
2. social determinants of health sensitivity analysis: male infertility status + age when first received male infertility diagnosis or vasectomy-related record + UC location + race + ethnicity + area deprivation index
3. hospital utilization sensitivity analysis: male infertility status + age when first received male infertility diagnosis or vasectomy-related record + UC location + number of visits in the EHR after the 6-month cutoff + number of months in the EHR after the 6-month cutoff

We found that patients with male infertility have a higher risk of receiving 11/13 diagnoses (with the exception of Other tests and Persons encountering health services in circumstances related to reproduction) after their male infertility diagnosis across all Cox regression analyses.

The findings can be found in Supplementary Data 21 and visualized in Supplementary Figure 13.

Reviewer #5 (Remarks to the Author):

In my view the authors have addressed most of the prior reviewer concerns. I believe the use of logistic regression for the post-diagnosis time window remains a methodological limitation, especially given the implied causal interpretation in certain parts of the manuscript. The authors have softened the language but have not changed the analytic approach.

In my reading, I believe this analytic approach could conflate directionality and does not account for variable follow-up time — which is crucial when asserting that downstream comorbidities might be outcomes of MI. A time-to-event model (e.g., Cox regression) or Poisson regression with offset for person-time would have been more appropriate. Sensitivity analyses are helpful but do not adequately describe how they handle censoring or missing diagnoses outside the healthcare system

Thank you for the feedback. We agree that it is essential to use time-to-event models when suggesting that downstream comorbidities could be outcomes of male infertility. Thus, we ran Cox proportional hazards models on the 13 diagnoses found to be significant after the 6-month cutoff at the University of California for the primary, social determinants of health, and hospital utilization logistic regression analyses. Similar to the logistic regression analyses, we ran 3 Cox regression analyses for each diagnosis by adjusting for the following covariates:

1. primary analysis: male infertility status + age when first received male infertility diagnosis or vasectomy-related record + UC location
2. social determinants of health sensitivity analysis: male infertility status + age when first received male infertility diagnosis or vasectomy-related record + UC location + race + ethnicity + area deprivation index
3. hospital utilization sensitivity analysis: male infertility status + age when first received male infertility diagnosis or vasectomy-related record + UC location + number of visits in the EHR after the 6-month cutoff + number of months in the EHR after the 6-month cutoff

We found that patients with male infertility have a higher risk of receiving 11/13 diagnoses (with the exception of Other tests and Persons encountering health services in circumstances related to reproduction) after their male infertility diagnosis across all Cox regression analyses.

The findings can be found in Supplementary Data 21 and visualized in Supplementary Figure 13.

Additional minor comments:

Use of Vasectomy Patients as Controls - While well justified (fertile comparator), there remains potential bias due to different healthcare-seeking behaviors.

Thank you for pointing this out. We have added this as a limitation in the Discussion section on page 16, lines 675-677:

While our sensitivity analyses try to account for this, they likely do not account for all potential bias, such as potential differences in the health-seeking behaviors of MI patients and vasectomy patients.

Multiple testing - The authors use Benjamini-Hochberg correction, but no power analysis or estimate of false positives from correlated phenotypes is provided.

Thank you for the feedback. We did not perform a power analysis. Thus, it is possible that there is reduced sensitivity that may lead to false negatives. We also did not estimate false positives from correlated phenotypes. These are limitations that we now highlight in the Discussion section on page 16, lines 687-688:

Another limitation is that we did not perform a power analysis; thus, some of our findings may be false negatives. We also did not estimate false positives from correlated diagnoses.

Reviewer #4 (Remarks to the Author):

The authors have addressed previous comments on methodology by adding analyses based on Cox regression. I think this is useful, but I agree with another reviewer that the design disregarding time should be mentioned as a limitation. Also a few more details on the Cox regression would be helpful, in particular how time was handled both with respect to the study design and the omission of the first 6 months.

Thank you for the feedback. We acknowledge that disregarding time in the logistic regression analyses is a limitation and have added the following to the Discussion on page 16, lines 695-700:

First, the logistic regression study design and its lack of accounting for time limits interpretation of findings, in particular the directionality of associations. Thus, longitudinal analyses are required to assess directionality. To address this, we used Cox proportional hazards models to further assess the relationship between male infertility and the 13 diagnoses found to be associated with male infertility for UC patients after the 6-month cutoff.

We also elaborated in the Methods how time was handled in the Cox proportional hazards models (page 6, lines 251-253 and underlined below):

Cox Proportional Hazards Models

We used Cox proportional hazards models to assess whether patients with an MI diagnosis (cases) had a higher risk of receiving MI-associated diagnoses relative to patients with a vasectomy-related record (controls). We focused on diagnoses that were first received beyond 6 months after a patient's first MI diagnosis and that were significantly associated with MI at UC for all 3 logistic regression analyses for a total of 13 diagnoses. For each analysis, a given diagnosis must be diagnosed in at least 1 case and 1 control patient. Three analyses were performed for diagnoses first diagnosed beyond 6 months after patients' first MI diagnosis or vasectomy-related record:

1. primary analysis: MI ~ estimated age + UC location
2. SDoH sensitivity analysis: MI ~ estimated age + UC location + race + ethnicity + ADI
3. hospital utilization sensitivity analysis: MI ~ estimated age + UC location + number of visits beyond 6 months after patients' first MI diagnosis or vasectomy-related record + months in the EHR

beyond 6 months after patients' first MI diagnosis or vasectomy-related record

EHR cutoff date was used to measure time-to-event for patients who did not receive a given diagnosis of interest; this date corresponds to June 30, 2023, the date the EHR was last refreshed for this study. Time-to-event corresponds to the number of days between patients' first male infertility diagnosis (cases) or vasectomy-related record (controls) and patients' first diagnosis of the diagnosis of interest.

Additionally, the schematic of the Cox proportional hazards model study design, which was previously Supplementary Figure 12, is now Figure 7.

Reviewer #5 (Remarks to the Author):

The authors have sufficiently addressed my prior comments but I think the Cox analysis should be featured in the main results rather than the supplement (due to the concerns with causality etc. for the logistic regression).

Thank you for the feedback. We added Table 3 to the main manuscript, which highlights the hazard ratio, 95% confidence interval bounds, standard error, statistic, and p-value associated with having male infertility for the Cox regression primary analysis. We also include the study design for the Cox proportional hazards models, which was previously Supplementary Figure 12, as Figure 7 in the main manuscript.